# Plasma exosomes from individuals with type 2 diabetes drive breast cancer aggression in patient-derived organoids
Christina S. Ennis[1], Michael Seen [1], Andrew Chen[2,3], Heejoo Kang[1], Adrian Ilinski[4], Kiana Mahdaviani[4], Naomi Y. Ko [4], Stefano Monti[2,3,5] & Gerald V. Denis [1,4,6] ✉

Women with obesity-driven type 2 diabetes (T2D) face worse breast cancer outcomes, yet metabolic status does not fully inform current standards of care. We previously identified plasma exosomes as key drivers of tumor progression; however, their effect on immune cells within the tumor microenvironment (TME) remains unclear. Using a novel patient-derived organoid (PDO) system that preserves native tumor-infiltrating lymphocytes (TILs), we show that T2D plasma exosomes induce a 13.6-fold expansion of immunosuppressive TILs relative to nondiabetic controls. This immune dysfunction may promote micrometastatic survival and resistance to checkpoint blockade, a known issue in T2D cancer patients. Tumor-intrinsic analysis revealed a 1.5-fold increase in intratumoral heterogeneity and 2.3-fold upregulation of aggressive signaling networks. These findings reveal how T2D-associated metabolic dysregulation alters tumor–immune crosstalk through previously underappreciated exosomal signaling, impairing antitumor immunity and accelerating progression. Understanding these dynamics could inform tailored therapies for this high-risk, underserved patient population.

Metabolic disorders, particularly Type 2 diabetes (T2D), raise concerns for the management of breast cancer. Population studies have shown that women with T2D have 40% increased risk of developing breast cancer and 74% increase in overall mortality compared to non-diabetic (ND) women[1]. The alarming impact of this comorbidity underscores a critical gap in our understanding of the cellular and molecular mechanisms linking these diseases. Recently, investigations into the immune phenotype of T2D have described subclinical chronic inflammation that induces insulin resistance and metabolic dysfunction that accompany the disease[2]. This inflammatory milieu ultimately leads to T cell dysfunction[3], posing a significant clinical challenge for breast cancer patients. Cancer cells have adaptations that allow them to exploit this dysfunction, reprogramming immune cells to evade detection and clearance[4]. Traditional molecular diagnostics fall short in capturing these profound differences seen in the tumor microenvironment (TME) of T2D[1], highlighting an urgent need for improved diagnostic and therapeutic approaches. Here, we use novel models to focus on unique features of this TME to explore mechanisms that may contribute significantly to breast cancer progression in T2D.

Within the breast TME, exosomes have emerged as key mediators of cellular communication. Initially dismissed as mere cellular disposal systems, these nano-sized extracellular vesicles are now recognized for their crucial roles in transferring bioactive molecules, such as proteins, lipids, and nucleic acids, between cells that influence tumor behavior and progression[5]. Our group and others have recently highlighted the critical role of exosomes in linking metabolic dysregulation with cancer pathophysiology[5–7]. We demonstrated that exosomes isolated from T2D patient adipocytes promote a more aggressive tumor phenotype in cellular models of breast cancer, increasing epithelial-to-mesenchymal transition (EMT) and cancer stem-like cell (CSC) formation, compared to ND controls[8]. Similarly, plasma-derived exosomes from T2D patients enhanced EMT and CSC traits in prostate cancer cell lines[9]. Furthermore, T2D exosomes carry distinctive biomarkers that could serve as diagnostic or prognostic tools[5,9], and offer new insights into the intercellular communication of the TME that drives breast cancer progression in metabolic disorders. However, these studies have yet to fully elucidate how exosomal treatment affects other cell types of the TME, and how this might exacerbate overall tumor aggressiveness.

[1]Cancer Center, Boston University Chobanian and Avedisian School of Medicine, Boston, MA, USA. [2]Department of Medicine, Computational Biomedicine Section, Boston University Chobanian and Avedisian School of Medicine, Boston, MA, USA. [3]Bioinformatics Program, Faculty in Computing and Data Science, Boston University, Boston, MA, USA. [4]Section of Hematology and Medical Oncology, Boston Medical Center, Boston, MA, USA. [5]Department of Biostatistics, Boston University School of Public Health, Boston, MA, USA. [6]Department of Pharmacology and Experimental Therapeutics, Boston University Chobanian and Avedisian School of Medicine, Boston, MA, USA. ✉e-mail: gdenis@bu.edu

Accurate modeling of the TME presents a challenge due to its intricate and dynamic cellular interactions. Recent advances in three-dimensional in vitro systems, such as patient-derived organoids (PDOs), have started to bridge this gap by preserving the heterogeneity and architecture of the originating tumors[10,11]. Leveraging this model, coupled with advancements in single-cell RNA sequencing—although sparingly used in breast cancer PDO research—have been instrumental in detailing the landscape of intratumoral heterogeneity preserved in PDOs[12]. However, a significant limitation has been the attrition and eventual elimination of the immune fraction[13]. Our novel approach enhances standard breast cancer PDO models by successfully preserving and comprehensively profiling native, primary breast tumor-infiltrating lymphocytes (TILs), for the first time to our knowledge. This advancement is a significant stride in breast cancer research, as it allows for a more accurate study of the immune-tumor interactions within the TME.

Here, we explore how exosomes derived from T2D patient plasma increase the aggressiveness of breast cancer. We hypothesize that T2D plasma-derived exosomes promote oncogenic processes, including EMT, CSC formation, and immune evasion, within the PDO model. Elucidating these interactions will refine our understanding of the role of T2D in breast cancer progression, offering novel insights that could fundamentally alter clinical approaches to breast cancer treatment.

## Results

### Single-cell RNA sequencing reveals cellular landscape of PDOs

To elucidate the cellular architecture and explore the effects of exosomal communication in breast cancer, we developed a robust in vitro model using PDOs generated from estrogen receptor-positive breast tumor resection samples from three ND patients (Fig. 1A and Supplementary Data 1). Briefly, tumor tissues were mechanically and enzymatically digested with

collagenase, and the resulting cell clusters were embedded in a basement membrane extract (BME) and cultured with a cocktail of growth factors to support organoid formation and maintain native TME populations and their neighborhood architecture. PDOs were then treated with plasma-derived exosomes from two carefully selected patients to minimize experimental variability: an ND individual (NDexo-PDOs) and a patient with T2D (T2Dexo-PDOs, glycated hemoglobin (HbA1c) 9.3%). These donors were chosen based on a preliminary screen of 15 patients, selecting those that would maximize the detection of functional differences in TME behavior. This approach ensured consistent experimental conditions across different PDO batches and facilitated a focused examination of how metabolic status influences TME dynamics. Plasma exosomes were isolated from EDTA-anticoagulated, peripheral venous blood, and quantified to deliver 5000 particles per PDO cell. A control group of untreated PDOs (UT-PDOs) was maintained to establish a baseline for comparison.

After stringent quality control (see "Methods"), we obtained transcriptomes of 24,599 single cells across three independently generated PDO lines each under three conditions, UT, NDexo, and T2Dexo subjected to exosome treatment (Fig. 1B–D, 318: UT—$n = 2007$, NDexo—$n = 2384$, T2Dexo—$n = 2562$; 377: UT—$n = 5396$, NDexo—$n = 6915$, T2Dexo—$n = 4953$; 409: UT—$n = 164$, NDexo—$n = 124$, T2Dexo—$n = 94$). The samples were integrated computationally using single-cell variational inference (scVI)[14] to address interpatient variability of tumor cell fractions (Fig. S1A, B). Copy number alterations were used to distinguish neoplastic from normal cells, leveraging immune cells also captured within PDOs (Figs. 1E and S2A–C). Tumor cell identities were validated by comparing gene expression profiles of breast cancer-specific gene sets from the Catalogue of Somatic Mutations in Cancer (COSMIC) Cancer Gene Census (CGC)[15] and gene sets related to mammary epithelial proliferation[16] (Fig. S1C).

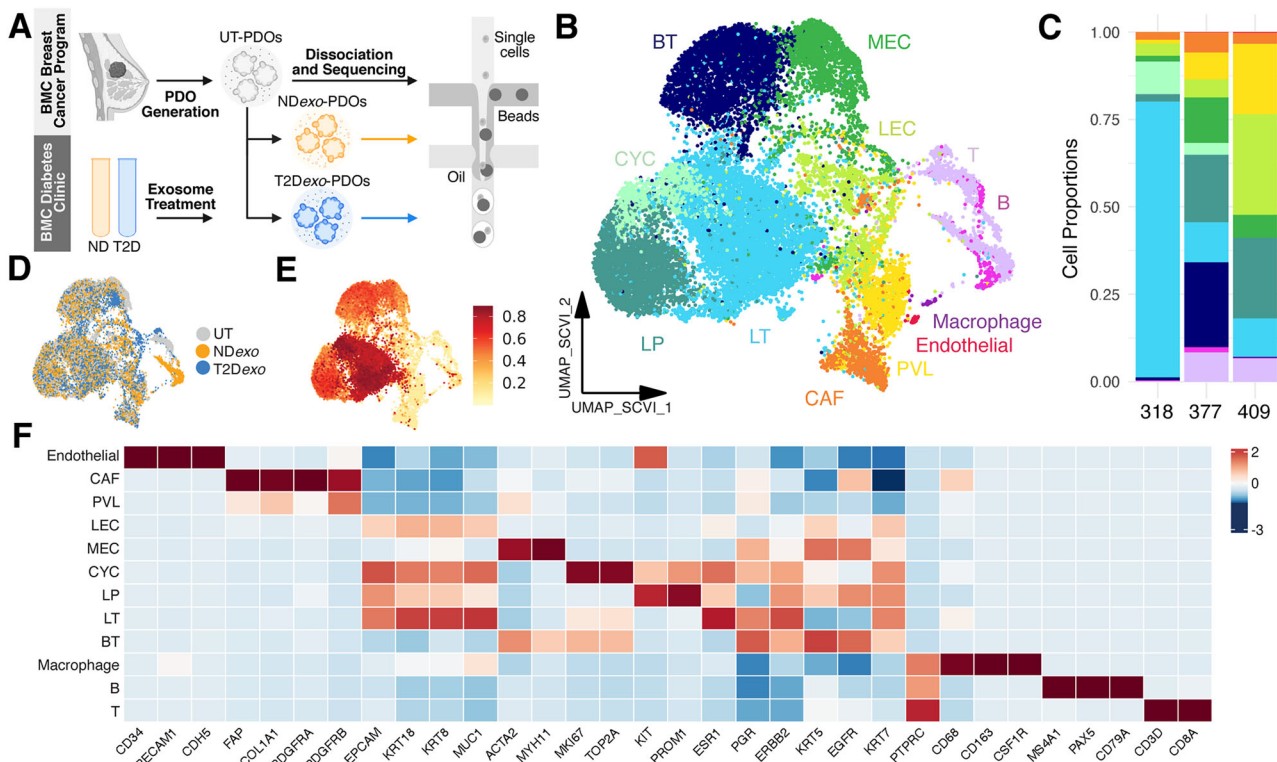

**Fig. 1 | Cellular characterization of PDOs. A** Illustration of scRNAseq workflow in PDO model of breast cancer TME[77]. **B** scVI-integrated UMAP visualization of PDOs colored by broad cell type annotation. $N = 24,599$ from three patients. BT ($n = 4226$) denotes basal-like tumor cells; CAF ($n = 1149$) denotes cancer-associated fibroblasts; CYC ($n = 1223$) denotes cycling epithelial cells; LEC ($n = 1254$) denotes normal luminal epithelial cells; LP ($n = 3574$) denote luminal progenitor-like cells; LT ($n = 7514$) denotes luminal-like tumor cells; MEC ($n = 2382$) denotes normal myoepithelial cells; PVL ($n = 1487$) denotes perivascular-like cells. **C** Relative proportion of cell types in each patient highlighting conservation of major lineages. Colors denote the same cell types in (**A**). **D, E** scVI-integrated UMAP colored by treatment (**D**) and malignancy score as calculated from genomic instability (**E**). **F** Heatmap showing expression of canonical marker genes per lineage.

Unsupervised uniform manifold approximation and projection (UMAP)-clustering revealed 17 clusters, which were further classified via automatic cell annotation[17,18] and established lineage-specific marker genes (Figs. 1F and S1D and Supplementary Data 2). Notably, many crucial cell types were abundantly present across all patient samples (Fig. 1C). We identified 20,173 epithelial cells, which accounted for the majority of sequenced cells. We identified three major stromal cell types (Fig. 1B), including cancer-associated fibroblasts (CAFS; *FAP, COL1A1*), perivascular-like cells (PVL; *PDGFRA, PDGFRB*), and endothelial cells (*PECAM1, CD34*). Critically, we identified distinct clusters relating to various immune cell types, including T cells (*CD3E, CD8A*), B cells (*MS4A1, PAX5*), and macrophages (*CD68, CD163*).

While PDO318 showed lower immune cell abundance compared to PDO377 and PDO409, the variation observed across the three PDOs likely reflects real interpatient differences in tumor-immune composition—a feature well-documented in breast cancer[18–20]. To contextualize these findings, we benchmarked our data against our recently assembled breast cancer single cell atlas[21], confirming that the cellular proportions in our PDOs fall within the spectrum observed in primary breast tumors (Fig. S1E). To control for variability across PDO lines, we explicitly model patient batch as a latent variable in our statistical framework. This approach, in which tumor, stromal, and immune populations are preserved in PDOs, represents a substantial advancement over traditional methods, which fail to maintain native breast TIL populations, and provides a comprehensive platform for studying immune-tumor interactions in breast cancer.

## T2D*exo*-PDOs exhibit aggressive phenotypes, compared to ND*exo*-PDOs

To elucidate the distinct transcriptional impacts of T2D-exosomal communication, we conducted gene set enrichment analysis (GSEA)[22] on differentially expressed genes across all cell populations (Supplementary Data 3). All changes we report were statistically significant. T2D*exo*-PDOs showed 2.1-fold upregulation of pathways related to EMT, invasiveness, and stemness compared to ND controls (Fig. 2A, B). This fold change was based on the normalized enrichment score from differential expression analysis. These hallmark processes are associated with aggressive tumor phenotypes and metastatic potential, consistent with our previous findings that T2D exosomes promote tumorigenic reprogramming in breast cancer models[8].

Validation of these findings was further supported by the upregulation of breast cancer genes listed in the COSMIC CGC and canonical CSC markers within the T2D*exo*-treated epithelial cell fraction compared to ND*exo*-PDOs (Fig. 2C). Top differentially expressed genes include well known drivers of aggressiveness and stemness (Fig. 2D and Supplementary Data 4; *NOTCH1, MMP14*). These results align with our prior PCR-validated data[8], strengthening the evidence that T2D exosomes promote aggressive cancer phenotypes. Morphological assessments of PDOs supported these findings, with T2D*exo*-PDOs displaying pronounced budding (Fig. 2E, F and Supplementary Data 5), a morphological indicator of increased metastatic capacity[23]. Such findings are consistent with our previous reports of T2D exosomes promoting migration and morphological changes in other cellular models[8,9,24], further reinforcing their role in enhancing tumor aggressiveness.

Conversely, ND*exo*-PDOs demonstrated 3.6-fold activation of T cell mediated immune responses (Fig. 2A, C, D and Supplementary Data 3). Specifically, pathways related to antigen binding and presentation, along with cytotoxic T cell signaling, were upregulated 3.9-fold and 3.3-fold, respectively, in ND*exo* relative to T2D*exo*-PDOs, indicating enhanced tumor immunogenicity and immune system recognition.

To evaluate the broader relevance of these findings, we generated a composite signature representing the net transcriptional imprint of diabetes-associated exosomal signaling. We applied this signature to two independent breast cancer cohorts—the Cancer Genome Atlas (TCGA)[25] and the Molecular Taxonomy of Breast Cancer International Consortium (METABRIC)[26]—and assessed its association with overall survival, adjusting for age, proliferation score, inflammation score, and PAM50 subtype.

Strikingly, this signature was associated with significantly worse survival outcomes in both datasets (Fig. 2G and Supplementary Data 6; TCGA: HR = 2.23, $p < 0.002$; METABRIC: HR = 2.06, $p < 1.2e-05$), underscoring its clinical relevance and generalizability beyond our in vitro system.

Functional annotation further revealed distinct biological programs: the T2D-like gene set was enriched for pro-tumorigenic pathways, including genes related to metabolism (26.5%), transcription/signaling (19.1%), invasion/EMT/proliferation (16.2%), ECM remodeling (11.8%), and stemness/development (7.4%). Only 5.9% of T2D-like genes were immune-related, half of which were suppressive. In contrast, the ND-like gene set was 91.4% immune-enriched, with dominant roles in T cell activation (30.6%), leukocyte migration (18.8%), and T cell development (16.5%), indicative of an antitumor immune phenotype.

These results reinforce epidemiologic findings that T2D is associated with worse breast cancer outcomes[27,28] and demonstrate that diabetes-linked exosomal programs can be linked to clinically meaningful survival differences. While TCGA and METABRIC do not include metabolic status metadata, this limitation highlights the unique value of our model in systematically linking systemic metabolic disease to tumor-intrinsic transcriptional states.

## Exosomes from T2D plasma increase intratumoral heterogeneity

Given the predominance of epithelial cells in our dataset, we next analyzed epithelial lineage-specific responses within T2D*exo*-PDOs versus ND*exo*-PDOs. We identified 11 distinct cell clusters (Fig. 3A, B and Supplementary Data 7), representing typical epithelial cell types, including normal luminal epithelial (LEC; *KRT18*), myoepithelial (MEC; *TAGLN, KRT14*), and cycling cells (CYC; *MKI67, TOP2A*).

Malignant cells were further classified into two distinct groups: luminal-like and basal-like. The luminal-like group consisted of five clusters (LT1-5), defined by *ESR1* positivity and expression of *KRT19*, whereas the basal-like group comprised two clusters (BT1-2) characterized by *ESR1* negativity and expression of canonical basal markers (*KRT5, VIM*). Despite all patients being clinically diagnosed as estrogen receptor-positive, our data, consistent with other studies[18], highlighted subtype mixing within individual tumors that was faithfully recapitulated in PDOs. In addition to these clearly malignant clusters, we identified a population of luminal progenitor-like cells (LP; *KIT, ELF5*), which are considered the presumptive cells of origin for breast cancer and are indicative of cancer stemness[18].

In the luminal-like group, we identified five subtypes (Supplementary Data 7). Clusters LT1 and LT2 showed high expression of genes associated with energy metabolism and oxidative stress (*MALAT1, NDUFC2, ELOB*). LT3 exhibited a heightened proliferation index, indicated by increased expression of *MKI67* and *TOP2A*, and genes marking breast epithelium (*ANKRD30A*). Importantly, whereas this cluster expressed proliferation-associated genes, it did not exhibit the same transcriptional profile as actively cycling cells, which clustered separately. This result suggests that LT3 represents a less proliferative, and potentially less aggressive, tumor subpopulation. LT4 was characterized by genes linked to increased immune infiltration (*HLA-A, B2M*). LT5 showed upregulation of genes associated with poor clinical outcomes (*MUC1, SERPINA3*). Within the basal-like fraction, most cells belonged to the BT1 cluster, marked by the expression of hallmark aggressive genes (*VIM, ITGA6*). Conversely, BT2 was characterized by genes associated with a protein synthesis phenotype (*RPS15*).

To probe how exosome treatment impacted luminal-like tumor cell niches, we conducted a differential proportion analysis of tumor clusters. We observed 1.5-fold expansion of the LT1 cluster, coupled with 2.2-fold contraction of LT3 in T2D*exo* compared to ND*exo*-PDOs (Figs. 3C and S3A, B). These fold changes were calculated by comparing the proportion of epithelial cells belonging to each cluster within each treatment group. LT1 marker gene analysis by GSEA revealed enrichment of gene sets related to transcriptional plasticity and vesicle trafficking, including RNA polymerase complex binding and phosphatidylinositol-3-phosphate biosynthesis (Fig. 3D and Supplementary Data 8), consistent with increased endocytic activity and potential responsiveness to exosome cargo.

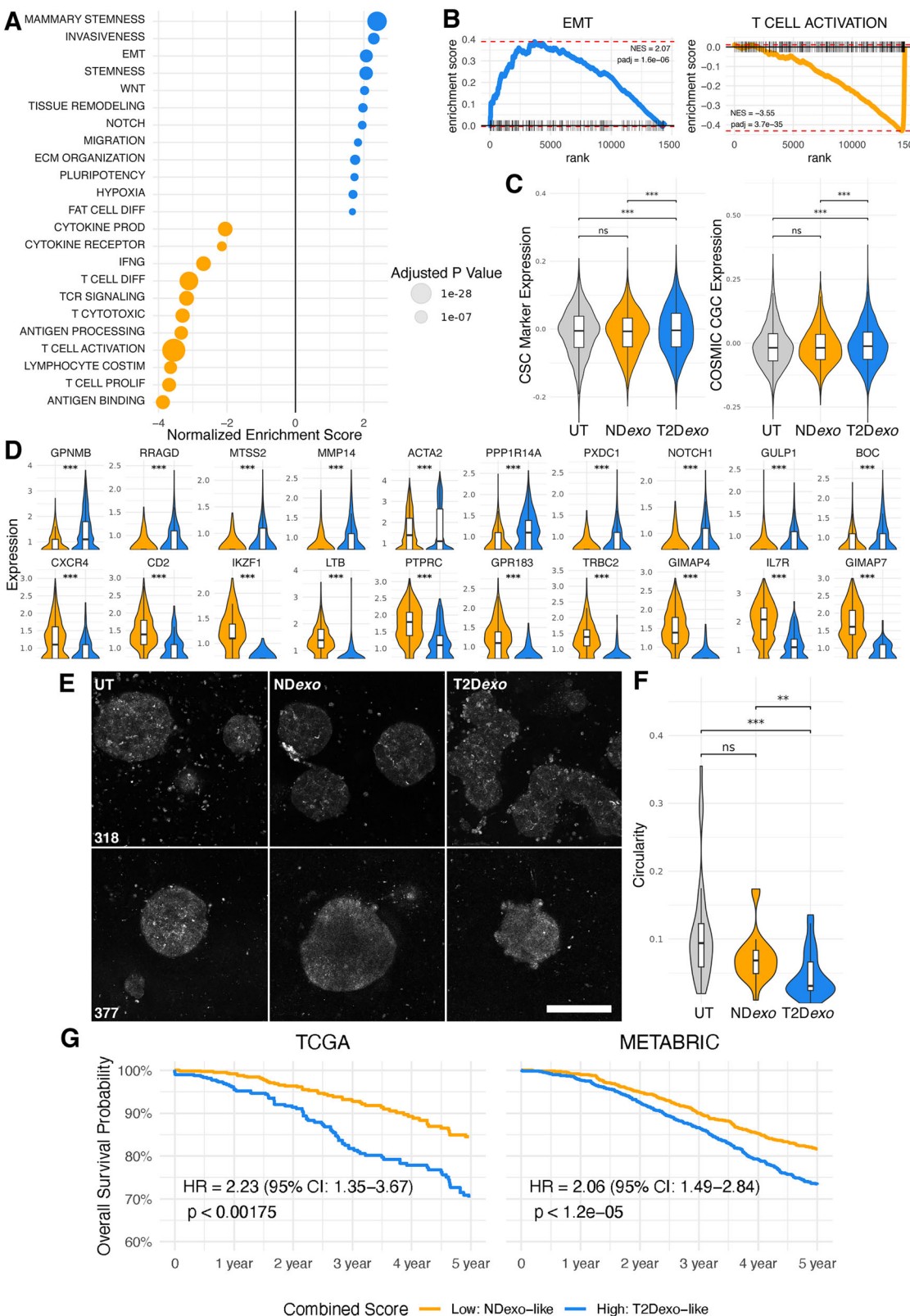

**Fig. 2 | Increased aggression in T2Dexo-PDOs compared to NDexo-PDOs.**
**A** Bubble plot summarizing statistically significant gene sets enriched in differentially expressed gene sets between T2Dexo-PDOs (blue) and NDexo-PDOs (orange) identified with GSEA. Size of dot is scaled to adjusted *p*-value. **B** GSEA enrichment plots for representative gene sets upregulated (top) and downregulated (bottom) in T2Dexo-PDOs compared to NDexo-PDOs. **C** Violin plots of COSMIC CGC (left) and canonical CSC gene expression (right). Significance calculated via linear mixed effects model, ***<0.001, **<0.01, *<0.05. **D** Violin plots of top differentially expressed genes. **E** Representative confocal images of UT (left), NDexo (middle), and T2Dexo (right) per patient. Scale bar = 100 μm. **F** Violin plots of calculated circularity of PDOs. Singlets were removed during thresholding. Significance calculated via pairwise Wilcoxon Rank Sums test with Bonferroni correction, ***<0.001, **<0.01, *<0.05. **G** Projection of composite gene signatures capturing diabetes-associated exosomal effects onto TCGA (left) and METABRIC (right) cohorts. Cox proportional hazards model adjusted for age, proliferation score, inflammation score, and molecular subtype.

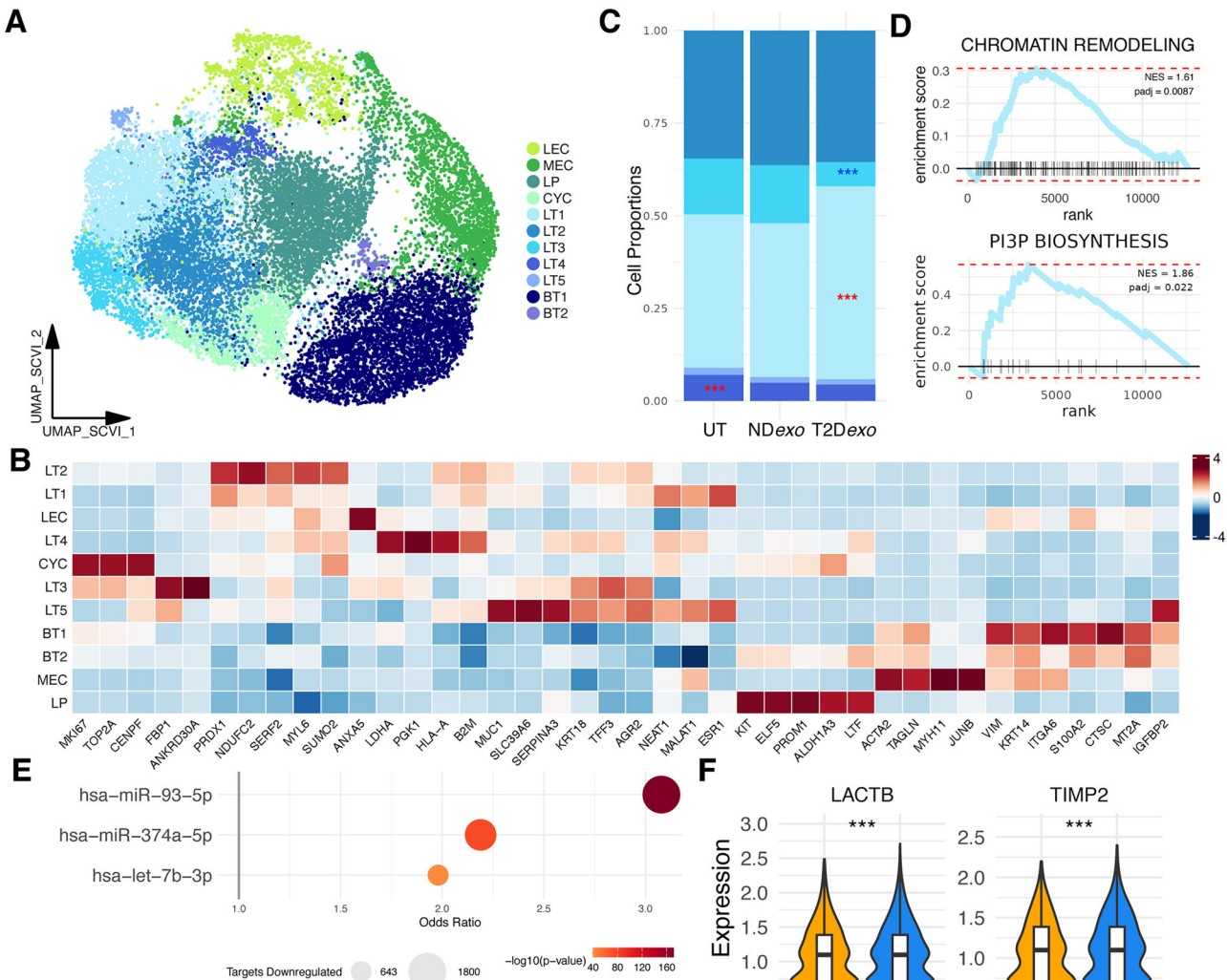

**Fig. 3 | Tumor population dynamics in T2Dexo-PDOs. A** UMAP visualization of reclustered scVI-integrated epithelial cells. BT1-2 (*n* = 4084; *n* = 142) denote basal-like tumor subclones; CYC (*n* = 1223) denotes cycling epithelial cells; LEC (*n* = 1254) denotes normal luminal epithelial cells; LP (*n* = 3574) denotes luminal progenitor-like cells; LT1-5 (*n* = 3397; *n* = 2669; *n* = 925; *n* = 398; *n* = 125) denote luminal-like tumor subclones; MEC (*n* = 2382) denotes normal myoepithelial cells. **B** Heatmap showing expression of canonical and top differentially expressed marker genes per subclone. **C** Relative proportion of luminal-like tumor subclones per treatment group. Red (blue) stars indicate significantly expanding (contracting) clusters. Significance calculated via binomial linear regression model, ***<0.001, **<0.01, *<0.05. **D** GSEA enrichment plots for representative gene sets upregulated in LT1. **E** Bubble plot summarizing enrichment of predicted and validated targets for individual T2Dexo-associated miRNAs among downregulated genes in LTs treated with T2Dexo vs. NDexo. Odds ratios from Fisher's exact test are shown for each miRNA; bubble size corresponds to the number of targets downregulated. **F** Violin plots showing expression of negative regulators of MMPs.

Surprisingly, we also observed enrichment of gene sets composed of predicted targets of miRNAs enriched in T2Dexo (Fig. S3C, D), including miR-374a, which we previously identified as significantly upregulated in T2D-derived exosomes[9]. While this might initially suggest upregulation of miRNA targets, we interpret this enrichment as a broader regulatory signature rather than direct target activation, potentially reflecting compensatory or indirect responses to miRNA exposure within LT1.

To directly evaluate the impact of T2Dexo-delivered miRNAs, we performed miRNA target enrichment across all LT clusters. Targets of validated T2D-associated miRNAs[9] (miR-374a-5p, miR-93-5p, let-7b-3p) were significantly enriched among genes downregulated in T2Dexo-treated LTs compared to NDexo (Fig. 3E, Fisher's exact test, $p < 2.2 \times 10^{-16}$; odds ratio = 3.6), consistent with canonical miRNA-mediated repression and providing strong evidence of functional miRNA activity in this context.

Consistent with prior findings[9,24], we observed upregulation of MMPs across all T2Dexo cells (Fig. 2D, *MMP2, MMP14*). In parallel, we found downregulation of key MMP inhibitors, including *TIMP2* and *LACTB*, specifically within T2Dexo LT clusters (Fig. 3F). Both inhibitors are

predicted or validated targets of T2D-enriched miRNAs (*TIMP2* by miR-93-5p; *LACTB* by miR-374a), suggesting that luminal-like tumor cells may be key sites of miRNA-driven repression that contributes to broader MMP activation. These findings support a model in which T2D-derived exosomes promote pro-invasive signaling across the TME.

**Exosome treatment accelerates tumor cell evolution**

To examine the evolutionary shifts induced by exosome treatment, we performed pseudotime analysis[29] on the luminal-like epithelial cell fraction in PDOs, excluding basal-like epithelial, immune, stromal, and cycling populations. Ordering these cells in pseudotime revealed a principal trajectory from normal LECs to more transformed states. Notably, cells from T2Dexo-PDOs were positioned significantly later in pseudotime compared to those from NDexo or UT-PDOs (Fig. 4A), indicating that T2D plasma-derived exosomes promote substantial evolution in tumor cell phenotypes. Overlaying tumor subclones onto the pseudotime trajectory (Fig. S4A) revealed that the LT1 subclone was positioned furthest along in pseudotime. This finding correlates with a higher proportion of LT1 cells in T2Dexo-PDOs, potentially explaining their advanced pseudotime positioning.

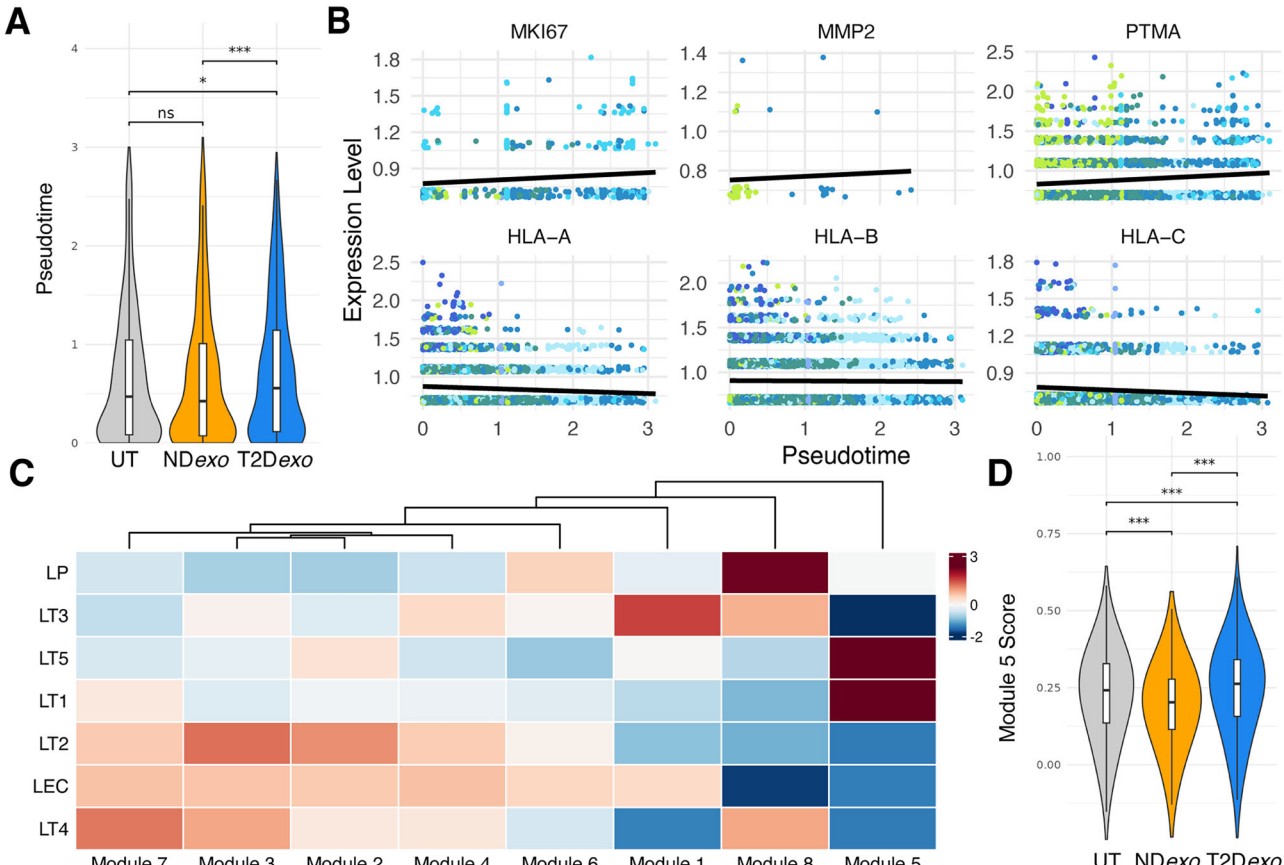

**Fig. 4 | Tumor cell evolution. A** Violin plots of pseudotime calculated for luminal-like epithelial cells per treatment group. Significance calculated via linear mixed effects model, ***<0.001, **<0.01, *<0.05. **B** Scatter plots of gene expression as a function of pseudotime. Dot color denotes epithelial cell subtype assigned to cells.

Black line showing linear regression of best fit. Genes determined via graph auto-correlation. **C** Heatmap of gene module expression in epithelial cell subtypes. **D** Violin plots of expression of Module 5 in luminal-like epithelial cells. Significance calculated via linear mixed effects model, ***<0.001, **<0.01, *<0.05.

To delineate the specific evolutionary paths captured by pseudotime, we performed differential gene expression using graph-autocorrelation. All changes we report were statistically significant. This analysis revealed upregulation of oncogenic genes (Supplementary Data 9; *MKI67, MMP2, PTMA*) along the trajectory (Fig. 4B). Conversely, major histocompatibility complex (MHC) class I molecules (*HLA-A, HLA-B, HLA-C*) were progressively downregulated, a common immune evasive strategy by cancer cells to thwart immune detection and clearance[30]. Grouping these differentially expressed genes into co-regulated modules resulted in eight distinct clusters (Fig. 4C). Importantly, module 5, which showed an increased expression through pseudotime (Fig. S4B, C and Supplementary Data 10), was upregulated in T2D*exo*-PDOs compared to ND*exo* or UT-PDOs (Fig. 4D). Functional enrichment analysis using the Database for Annotation, Visualization, and Integrated Discovery (DAVID) resource[31] revealed that this module is involved in EMT (*CTNNB1, TRIM28, BRD2*), RNA splicing (*SNRNP70, SRSF4*), and aerobic metabolism (*ATP5F1A, MT-ND5*) which are well known to be perturbed in T2D[1,32]. These findings are consistent with our previous reports where treatment of MCF7 cells with T2D-derived exosomes induced shifts in metabolic pathways[8].

**T2D*exo*-PDOs display enhanced immune evasion through T cell dysfunction**

Given the significant downregulation of T cell signaling, we conducted a focused analysis of the immune cell compartment within T2D*exo*-PDOs compared to ND*exo*-PDOs. We identified 10 distinct cell clusters representing 1760 various immune cells (Fig. 5A, B and Supplementary Data 11) upon reclustering, including T cells (*CD3E, TRAC*), B cells (*MS4A1, CD79A*), plasma cells (*MZB1, SDC1*), and macrophages (*CD68, CD163*).

T cells were further classified into six distinct subsets. The largest proportion consisted of memory T cells, including central memory (Tcm; *SELL, CCR7*) and effector memory T cells (Tem; *SELL-, CCR7-*). We also identified clear populations of active anti-tumorigenic T cell states, including effector T (Teff; *CD8A, GZMB*) and T helper 1 cells (Th1; *CD4, IFNG*). We also found a notable population of mucosal-associated invariant T cells (MAIT; *KLRB1, IL18R1*), an unconventional subset recognized for their innate-like effector functions and potential importance in antitumor immunity[33]. In addition to these well-defined T cell subsets, we identified a conserved cluster (ChopT) characterized by the expression of *DDIT3*, which encodes the transcription factor CHOP (Fig. S5A–C). CHOP is a downstream sensor of severe endoplasmic reticulum (ER) stress, and acts as a major negative regulator of both CD8+ and CD4 + T cell effector function[34].

We next explored how exosome treatment alters native TIL function by performing differential cell proportion analysis (Supplementary Data 12). We saw 13.6-fold expansion of ChopT cells, coupled with 5.3-fold and 6.5-fold contraction of Tcm and MAIT, respectively, based on immune cell proportions within each treatment group (Fig. 5C). Further, GSEA analysis restricted to immune cells highlighted enhanced anti-inflammatory signaling via interleukin-10 (IL-10) and decreased proinflammatory signaling via the tumor necrosis factor (TNF) superfamily (Figs. 5D and S5D and Supplementary Data 13)[30]. We also identified 1.5-fold and 1.8-fold upregulation of gene sets related to hypoxia and defective metabolism, respectively. Additionally, GSEA revealed upregulation of cytotoxic and effector responses, including antigen binding and T cell receptor (TCR) signaling, in ND*exo*-PDOs (Fig. 5D). Collectively, these data demonstrate a clear and distinct profile of immune evasion in T2D*exo* compared to ND*exo*-PDOs.

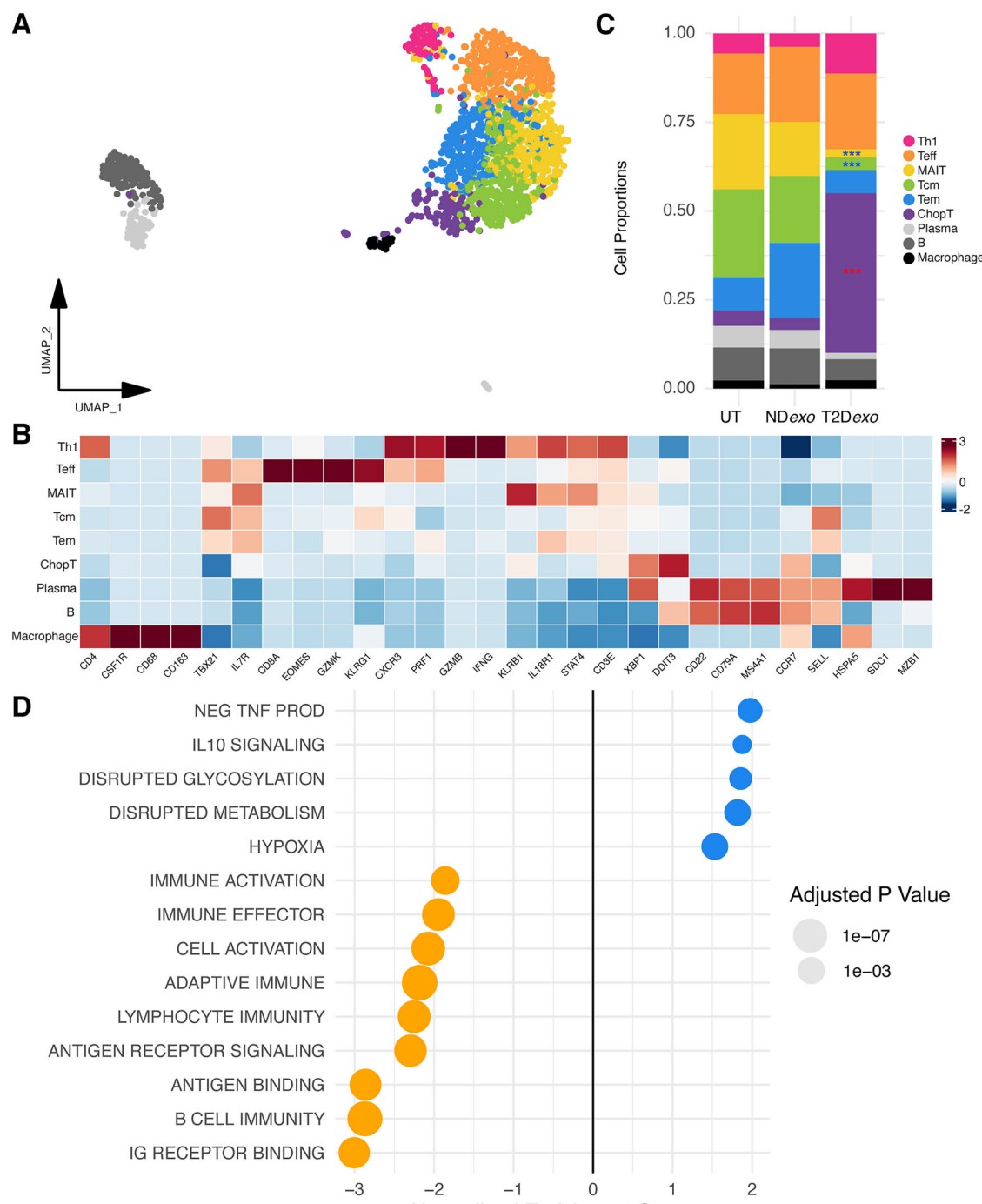

**Fig. 5 | Perturbed immune signaling in T2Dexo-PDOs. A** Unintegrated UMAP visualization of clustered immune cells. ChopT (*n* = 135) denote *DDIT3* + T cells; MAIT (*n* = 284) denote mucosal-associated invariant T cells; Tcm (*n* = 343) denotes central memory T cells; Tem (*n* = 274) denotes effector memory T cells; Teff (*n* = 374) denotes effector T cells; Th1 (*n* = 90) denote T helper 1 cells. **B** Relative proportion of immune compartment per treatment. Red (blue) stars indicate significantly expanding (contracting) clusters. Significance calculated via binomial linear regression model, \*\*\*<0.001, \*\*<0.01, \*<0.05. **C** Heatmap showing expression of canonical marker genes per immune cell state. **D** Bubble plot summarizing statistically significant gene sets enriched in differentially expressed genes between the immune compartments of T2Dexo-PDOs (blue) and NDexo-PDOs (orange) identified during gene set enrichment analysis (GSEA). Size of dot is scaled to adjusted *p*-value.

To further investigate the emergence of ER-stressed T cells in T2Dexo-PDOs, we performed pseudotime analysis on the T cell compartment (Fig. 6A). This approach allowed us to map activation and differentiation trajectories of TILs under the influence of T2D-derived exosomes. In NDexo-PDOs, CD8⁺ T cells followed canonical activation and differentiation pathways toward effector and memory states. In contrast, CD8⁺ T cells in T2Dexo-PDOs deviated sharply from this trajectory, branching instead into an alternative path marked by ER stress, metabolic dysfunction, and effector failure (Fig. 6B). This dysfunctional path was enriched for genes involved in the unfolded protein response (UPR), oxidative stress, and apoptosis—including high expression of *DDIT3* and *XBP1*—and was depleted for key regulators of T cell function and persistence, such as *IL7R*, *GZMK*, and *TBX21* (Fig. 6C–E and Supplementary Data 14). This finding is consistent with other reports of T-bet repression in ER-stressed T cells[34]. We

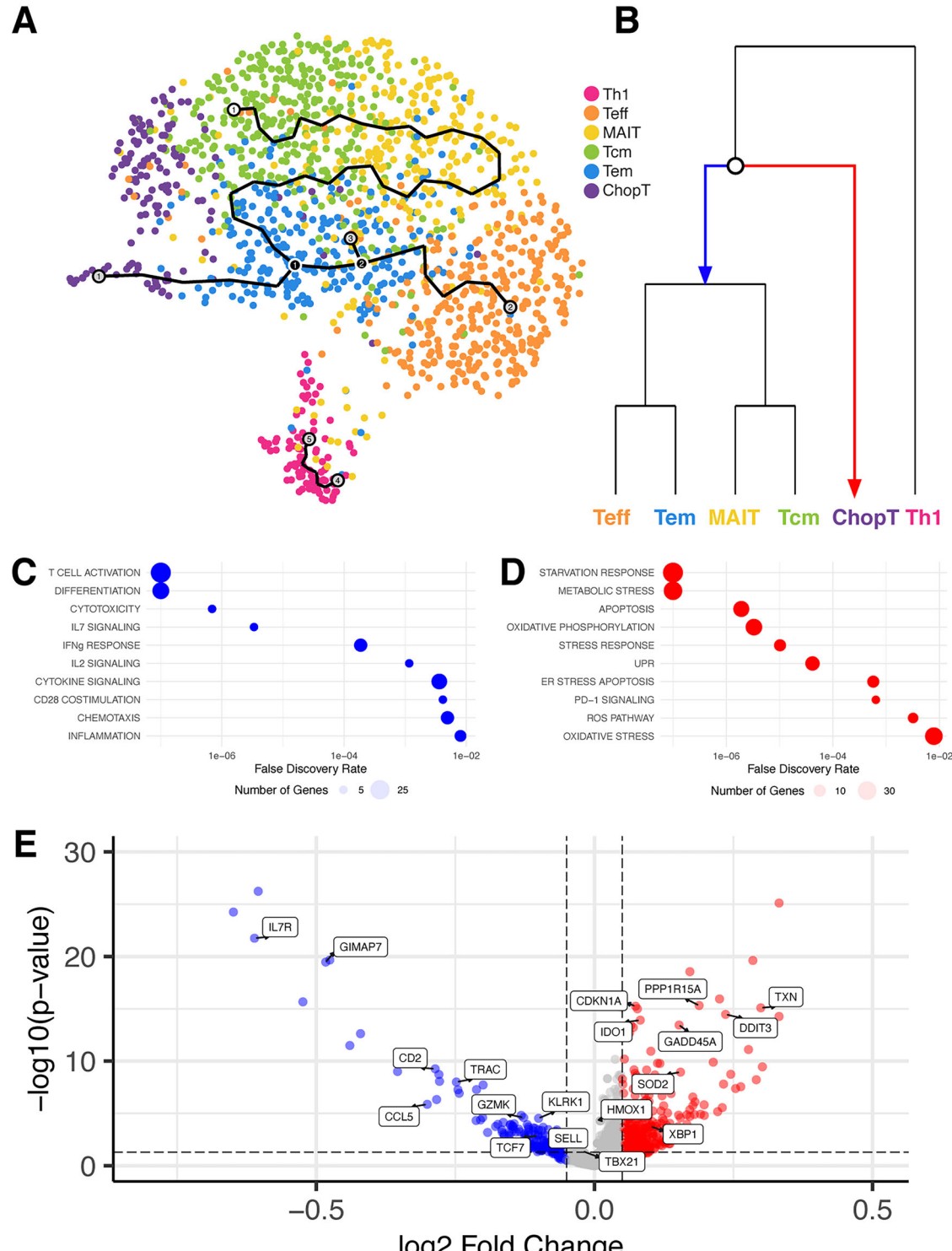

**Fig. 6 | Pseudotime analysis reveals divergent fates in the T cell compartment.**
**A** Trajectory inferred from T cells. **B** Phylogenetic tree showing branch point leading either toward typical development (blue) or toward ChopT phenotype (red). **C, D** Bubble plot summarizing statistically significant gene sets enriched along the trajectory toward proper development (**C**, blue) or toward the ChopT phenotype (**D**, red). Dot size reflects the number of genes detected per gene set. **E** Volcano plot of top differentially expressed genes at the branch point.

also observed upregulation of *IDO1*, a tryptophan-catabolizing immunoregulatory gene, further supporting the emergence of a checkpoint-independent suppressive program. These findings suggest that T2D-derived exosomes redirect T cell differentiation away from effector fates, instead driving a chronic dysfunction program underpinned by ER stress, metabolic reprogramming, and transcriptional suppression.

To independently validate these findings in a broader clinical context, we applied this ChopT signature to our recently assembled pan-breast cancer single-cell atlas[21]. We calculated per-cell enrichment scores for the ChopT program, along with canonical and curated T cell modules representing effector function, cytotoxicity, IFN-γ signaling, and stress-related pathways. Across this large and diverse clinical cohort, we

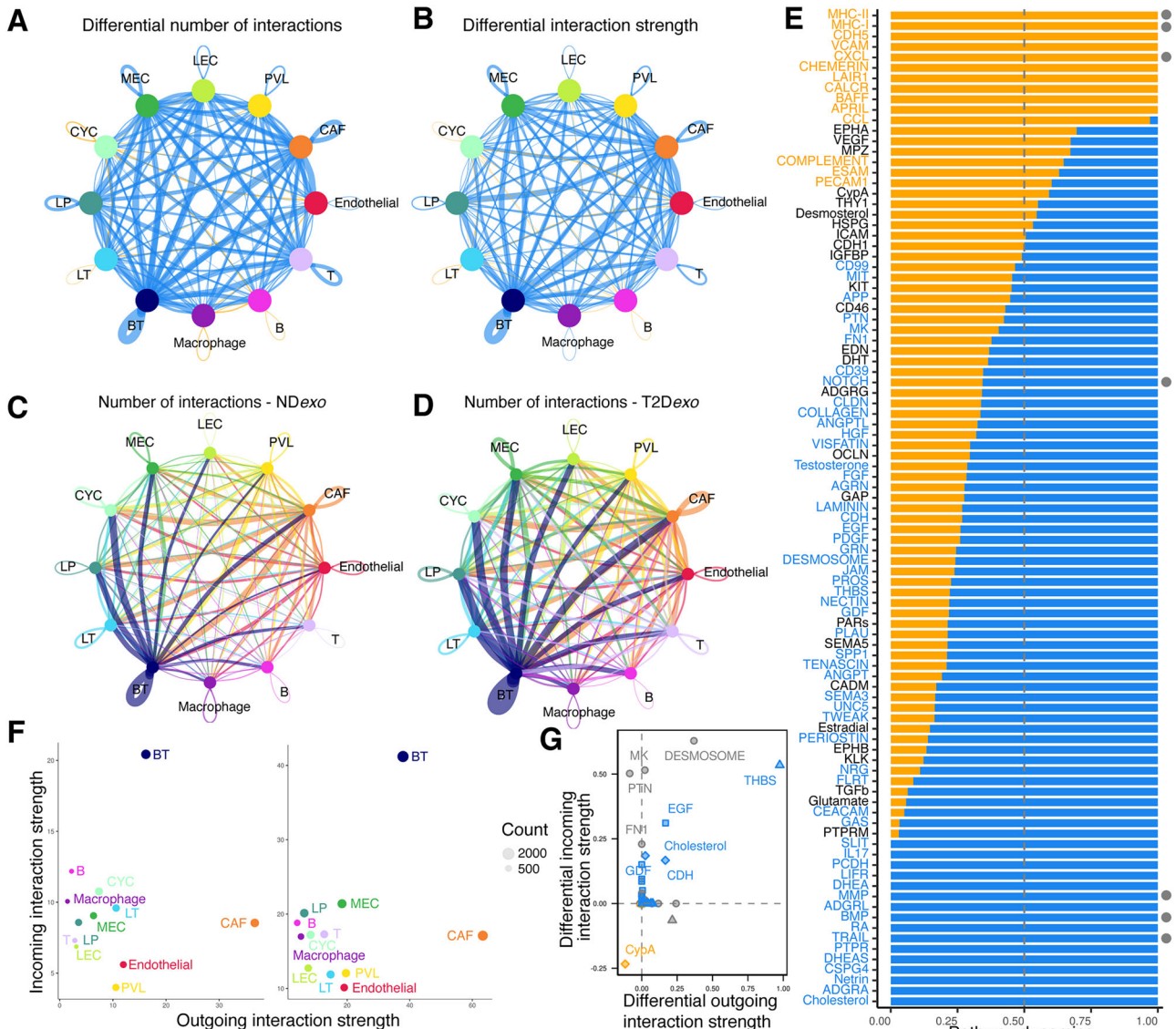

**Fig. 7 | T2Dexo modulates intercellular communication networks. A** Circle plot showing the number of interactions among different cell types. The blue (orange) colored edges represent increased (decreased) signaling in T2Dexo compared to NDexo-PDOs. **B** Circle plot showing the differential interaction strength among different cell types. The blue (orange) colored edges represent increased (decreased) interaction strength in T2Dexo compared to NDexo-PDOs. **C, D** Circle plot displaying the number of interactions and the strength of interactions between cell types in NDexo-PDOs (**C**) and T2Dexo-PDOs (**D**). The number of lines represents the number of interactions, and the thickness of lines is proportional to the strength of the interactions. **E** Stacked bar plot showing the overall information flow of each significant signaling pathway (*p* < 0.01). The vertical dashed line indicates the position where the sample accounts for 50% of the overall information flow. Label color is determined by a significantly larger contribution from T2Dexo (blue) or NDexo-PDOs (orange). Pathways equally important in both datasets are labeled in black. *P* value determined by paired Wilcoxon test according to CellChat. Grey dots indicate pathways highlighted within the main text. **F** Scatter plot of dominant senders and receivers for NDexo (left) and T2Dexo-PDOs (right). **G** Scatter plot of the signaling changes associated with T cells in T2Dexo (blue) or NDexo-PDOs (orange). Shared pathways are labeled in gray. Collagen and laminin pathways have been removed from plot for ease of visualization.

observed that ChopT-high cells consistently exhibited features of stress-induced dysfunction (Fig. S5E). Specifically, ChopT module scores showed positive correlations with stress-related circuits (e.g., $\rho = 0.34$ for hypoxia score; $\rho = 0.30$ for UPR score; $\rho = 0.24$ for ER stress apoptosis score; $p < 1e-10$), and negative correlations with key effector-related programs (e.g., $\rho = -0.24$ for effector score, $\rho = -0.14$ for cytotoxicity score, $\rho = -0.12$ for interferon-$\gamma$ signaling; $p < 1e-10$). These results confirm that the ChopT transcriptional program marks a reproducible, functionally impaired T cell state enriched for ER stress signatures, validating the PDO-derived observations in primary human tumors. They further support the interpretation that ER stress is a central driver of T cell dysfunction in T2D-influenced TMEs, potentially overriding classical checkpoint-based mechanisms.

## Exosomes modulate intercellular communication in T2Dexo-PDOs

To further characterize the immune evasive TME of T2Dexo-PDOs compared to NDexo-PDOs, we performed ligand-receptor interaction analysis[35] between all identified cell types (Figs. 7A–D, and S6A, B). Overall, 95 pathways relating to 1142 genes were involved in building the communication network, including 69 conserved pathways, 10 NDexo-specific and 16 T2Dexo-specific pathways (Fig. 7E and S6C, D and Supplementary Data 14). Here, we focused our analysis on a subset of biologically relevant pathways associated with immune regulation and tumor-promoting signaling. This analysis revealed upregulated NOTCH (e.g., JAG1—NOTCH1) and bone morphogenic protein (BMP; e.g., BMP7—[BMPR1B + BMPR2]) signaling in T2Dexo-PDOs, promoting stem-like

characteristics and resistance to apoptosis[36]. These findings are supported by our previous work, where we consistently observed downregulation of apoptosis in T2D exosome-treated models[8,24]. Additionally, T2D*exo*-PDOs exhibited enhanced extracellular matrix remodeling with heightened interactions involving MMPs (e.g., MMP2—([TGAV + ITGB3]) crucial for tumor cell invasion and migration[37]. Further, T2D*exo*-PDOs showed increased TNF-related apoptosis-inducing ligand (TRAIL) signaling (e.g., TNFSF10—TNFRSF10B), which has been linked to immunosuppression within the TME[38]. Conversely, ND*exo*-PDOs displayed upregulated communication involving MHC class-I, MHC class-II, and CXCL signaling, indicative of a more robust antitumor immune response[30]. All changes we report were statistically significant.

We observed a dense communication network in T2D*exo*-PDOs among basal-like tumor cells and CAFs with all other annotated cell types (Fig. 7F). Focusing specifically on differential T cell ligand-receptor interactions, T2D*exo*-PDOs demonstrated distinct signaling alterations, notably in the thrombospondin pathway (Figs. 7G and S7A, B). This pathway is known for its immunosuppressive action in T cells, promoting tumor escape mechanisms in breast cancer[39]. Notably, we have previously identified TSP5, a member of the thrombospondin family, as significantly enriched in T2D-derived exosomes compared to those from ND[8]. In contrast, ND*exo*-PDOs exhibited specific activity in the CypA pathway, which is known to significantly limit tumor growth and enhance antitumor immunity[40]. This ligand-receptor analysis further substantiates the role of T2D-derived exosomes in fostering a more aggressive and immune-suppressed TME.

## Discussion

T2D has long been established as a risk factor for poorer prognosis of breast cancer across numerous population studies[1]. Epidemiological data indicate that women with T2D have a higher incidence of breast cancer compared to ND and they often present with more aggressive tumor phenotypes[27]. Meta-analyses and large-cohort studies have reinforced these findings, suggesting that metabolic dysregulation in T2D may contribute to the pathogenesis and progression of breast cancer[41]. Much of the existing research has focused on tumor-intrinsic properties, including insulin-like growth factor signaling and excess glucose availability feeding tumor growth[1]. However, the role of the TME in this comorbidity has been largely overlooked. In this study, we aimed to investigate the impact of metabolic dysregulation in T2D on breast cancer aggressiveness and TME dynamics.

Exosomes have emerged as critical mediators of intercellular communication, enhancing cancer progression across various tumor types[5]. Of particular interest, plasma-derived exosomes from individuals with T2D have been shown to induce upregulation of gene sets associated with EMT, invasion, metastasis, and CSC formation in cancer cell lines[8,9]. Despite their significant influence on recipient cell behavior, it remains unclear how T2D-derived exosomes may alter niche dynamics within the TME. In this study, we extended our previous findings by utilizing PDOs to investigate exosomal signaling within the breast TME. By examining these communication networks, we sought to elucidate how exosomes contribute to breast cancer aggressiveness in the context of T2D. We used plasma-derived exosomes to simulate the influence of T2D-associated systemic factors on the TME. Although this approach does not entirely capture the complexity of PDOs from T2D samples, it provided a robust platform for exploring how T2D-derived exosomes contribute to a more aggressive and immunosuppressive TME.

Here, we describe a novel breast cancer PDO methodology that facilitates in vitro TME study by uniquely preserving native TILs and stroma. Whereas previous researchers have generated organoids that include primary TILs[42], these methods have not been established for breast cancer. Our approach, characterized by a short-term culture, offers advantages for experimental throughput and timely data acquisition that accelerates the pace of research and potential therapeutic development. By preserving native TIL populations, our model system allows for accurate modeling of human tumor-immune interactions, as opposed to traditional epithelial-only organoid models[43,44], or models that rely on the co-culture

reconstitution of peripheral blood[45]. Importantly, our method maintains a diverse array of endogenous immune cell types, including B cells and macrophages, alongside T cells, unlike reconstitution models that use clonally expanded or TCR-engineered populations[46].

In this study, we found that T2D*exo*-PDOs displayed enhanced EMT and CSC characteristics compared to ND*exo*-PDOs. Specifically, we observed upregulation of multiple gene sets related to these pathways, including NOTCH signaling, WNT signaling, EMT, migration, and multi-cancer invasiveness signatures (Fig. 2A, C, D). Furthermore, projecting the T2D*exo*-induced gene signature onto the TCGA and METABRIC datasets revealed a strong association with poor patient survival outcomes (Fig. 2G). Pseudotime analysis revealed distinct modules of co-regulated gene expression, with Module 5 marked by EMT genes (Fig. 4D). Ligand-receptor analysis further highlighted the influence of NOTCH signaling on intercellular communication in T2D*exo*-PDOs (Fig. 7E). Morphological assessments supported these findings, as T2D*exo*-PDOs displayed pronounced budding (Fig. 2E, F). The networks that we see perturbed are remarkably similar to what we have already published in multiple T2D exosome models, including breast cancer cell lines[8,9,24], suggesting a conserved mechanism of action of exosomes. These findings underscore the profound impact of metabolic dysregulation on breast cancer progression, highlighting the need for targeted therapeutic strategies that address the altered niche dynamics induced by T2D-derived exosomes.

In contrast, we observed suppression of immune function in T2D*exo*-PDOs compared to ND*exo*-PDOs, characterized by downregulation of gene sets involved in multiple steps of immune function, including cytotoxic signaling, activation, cytokine production, and proliferation (Fig. 2A, C). Restricting our analyses to the immune compartment, we further observed upregulation of IL-10 signaling and hypoxia signatures, alongside downregulation of immune activation and effector function programs in T2D*exo*-PDOs (Fig. 5D). A deeper analysis of the immune state dynamics revealed expansion of ChopT cells (Figs. 5A and S5A–C), a distinct and conserved T cell state indicative of severe ER stress. Critically, these cells emerged along a divergent pseudotime trajectory (Fig. 6A), bypassing canonical effector and memory differentiation pathways in favor of a transcriptional program marked by metabolic dysfunction, UPR activation, and loss of effector capacity. This reprogramming highlights a checkpoint-independent mechanism of T cell suppression, wherein ER stress fundamentally redirects differentiation toward a dysfunctional state.

Interestingly, our ligand-receptor analysis highlighted T2D*exo*-specific TRAIL signaling (Fig. 7E), which has been linked to the UPR[38]. Notably, thrombospondin signaling was also specifically active in T2D*exo*-T cells (Fig. 7G), aligning with our previous findings of TSP5 upregulation in T2D-derived exosomes compared to ND controls[8]. This finding highlights the significant role of thrombospondin family proteins in exosome effector function, influencing not only tumor cells but also the broader TME. Interestingly, we did not observe a significant change in Teff proportions, contrary to what might be expected in the pro-tumorigenic TME in T2D; instead, there was contraction of MAIT cells (Fig. 5C), consistent with reports of diminished MAIT frequency and functionality in adults with T2D, particularly in obese patients[47]. Collectively, these results suggest a coordinated pattern of immune evasion in T2D, driven by ER stress and altered cytokine signaling, likely shaped by the hypoxic and metabolically compromised conditions of the TME.

Contrary to our expectations, we did not observe upregulation of checkpoint ligands on T cells, which would have indicated classical exhaustion. Immune exhaustion is a well-characterized mechanism of dysfunction in cancer, marked by sustained expression of inhibitory receptors and progressive loss of effector function[4]. While exhaustion phenotypes are known to worsen in T2D and contribute to poor clinical outcomes[3], the immune profile observed here suggests a distinct, non-canonical pattern of T cell impairment.

Our data instead point to ER stress as a central driver of T cell dysfunction. CHOP (*DDIT3*), a downstream effector of the PERK–ATF4 axis of the UPR, has been shown to suppress CD8⁺ T cell function in solid tumors

independently of checkpoint signaling[48,49]. The IRE1α–XBP1 branch of the UPR is similarly implicated in regulating hypoxia-related gene expression[50] and inhibiting T cell infiltration and IFNγ production[51], while PERK–eIF2α signaling can directly impair T cell functionality and reduce tumor infiltration[52]. CHOP expression, in particular, has been consistently associated with immunosuppressive signaling and poor T cell effector function[34,53].

Given that T2D-derived exosomes robustly activate UPR pathways—as evidenced by elevated *DDIT3* and *XBP1* expression—they may drive T cells into a dysfunctional state prior to the onset of classical exhaustion. Notably, pharmacologic or genetic inhibition of CHOP has been shown to reinvigorate T cell activity[34], highlighting ER stress not only as a biomarker but as a potential barrier to effective immunity.

ER stress is also a key pathological mechanism of T2D, contributing to β-cell failure, insulin resistance, and chronic inflammation[54–57]. Our findings are among the first to mechanistically link ER-stressed T cells to the diabetes-associated TME, offering a new perspective on how comorbid metabolic disease reshapes antitumor immunity.

These results have direct therapeutic implications. Standard immunotherapies that rely on checkpoint inhibition may be ineffective in this context, where T cells are not classically exhausted but rather reprogrammed by ER stress. This result is consistent with existing literature reporting a diminished response to immunotherapy in T2D cancer patients[58,59]. Given the distinct immune landscape in breast cancer with comorbid T2D, therapeutic strategies must address the specific immunometabolic perturbations observed. Approaches that alleviate ER stress, modulate hypoxic responses, and restore T cell function may hold greater promise than checkpoint inhibitors alone.

While our study included a limited number of PDOs (*n* = 3) and plasma donors (*n* = 2), this focused design was intentional and necessary to mechanistically dissect the immunologic impact of T2D-derived exosomes under well-controlled conditions. Prior work from our group has demonstrated that exosomes from ND individuals exhibit high biological reproducibility across donors, with minimal inter-donor variability in gene expression profiles and consistent clustering with untreated controls[9]. In contrast, T2D is a biologically heterogeneous condition influenced by glycemic status, inflammation, comorbidities, and treatment exposures, all of which can modulate exosome composition and function. Rather than obscuring this complexity through overaggregation, we chose a defined T2D phenotype (see "Methods") to establish a tractable model for studying diabetes-driven tumor-immune interactions. Moreover, our prior studies demonstrate that T2D status exerts a more pronounced effect on immune-related pathways in the TME than other metabolic factors, such as cholesterol levels or waist-to-hip ratio[27,41,60–62], and these findings are further supported by extensive evidence linking T2D to immune dysfunction[63–65]. Together, these data underscore the rationale for focusing on T2D as a driver of immunologic remodeling in breast cancer and for anchoring this study in a rigorously controlled donor comparison. We are actively expanding our studies to include additional patients and breast cancer subtypes, particularly estrogen receptor–negative and triple-negative breast cancers, which are more prevalent in individuals with T2D[27]. Toward this goal, we are building a large-scale single-cell breast cancer atlas[21] to enable systematic investigation of the TME across a more diverse range of clinical and metabolic states. Additionally, while the present study did not examine how patient medications might influence exosome payloads, future work will explore how factors such as insulin sensitizers, weight loss agents, exercise, and diet impact exosome-mediated signaling. These efforts may uncover anti-metastatic or immune-restorative effects of metabolic interventions mediated by specific miRNAs and inform therapeutic strategies tailored to comorbid cancer populations.

Together, these findings enhance our understanding of dynamic interactions within the TME and offer new insights into the significance of novel exosomal communication on tumor biology. Our results suggest that metabolic status, particularly T2D, could be considered in the clinical management of estrogen receptor-positive breast cancer to improve therapeutic outcomes. Given the unique vulnerabilities of this population, we strongly advocate for the initiation of targeted clinical trials to explore the TME in these 120 million underserved and understudied patients. Addressing these gaps in knowledge and care could pave the way for more effective, personalized treatment strategies, ultimately improving survival and quality of life for this high-risk group.

## Materials and methods
### Patient sample collection
This study was conducted with approval from the Boston University (BU) Institutional Review Board (IRB), ensuring adherence to ethical standards. Informed consent was obtained from each participant prior to enrollment. All ethical regulations relevant to human research participants were followed. Breast tumor tissues were harvested from three patients with histologically confirmed estrogen receptor-positive breast cancer during resection surgery. None of the patients had received chemotherapy or radiotherapy prior to surgery, avoiding potential confounders in tumor biology. The collected tissues were processed immediately after resection. Samples were aseptically minced into approximately 1 mm$^3$ chunks using sterile scalpel blades, facilitating even freezing and subsequent thawing. Chunks were then cryopreserved in Cryostor® CS10 freezing media (StemCell, #100-1061), designed to minimize cryoinjury, at a controlled cooling rate of −1 °C per minute, and stored at −196 °C until further processing.

### Tissue digestion and organoid generation
PDOs were cultured from cryopreserved breast tumor samples following a modified version of previously published protocols[11,43,44]. A two-step enzymatic digestion process was implemented to ensure gentle dissociation while preserving vital stromal and immune subsets. Initially, tissues were incubated at 37 °C for 1 h under 200 rpm with 2 mg/mL collagenase III (StemCell, #07422) in a complete medium consisting of Hyclone DMEM/F/12 1:1 (Thermo Scientific, #SH30023.01) supplemented with 15 mM HEPES, 2.5 mM L-glutamine, 10 μg/mL gentamicin (Sigma, #G1272), 1% penicillin/streptomycin (Sigma, #516106), 2.5 μg/mL Amphotericin B (Fisher, #MT30003CF), 5 mM nicotinamide (Sigma, #N0636), 1.25 mM N-acetylcysteine (Sigma, #A9165), 1× B27 supplement (Fisher, 17-504-044), 250 ng/mL R-spondin 3 (Sigma, #SRP3323), 5 nM heregulin (StemCell, #78071), 5 ng/mL KGF (StemCell, #78046), 20 ng/mL FGF10 (StemCell, #78037), 5 ng/mL EGF (StemCell, #78006), 100 ng/mL Noggin (StemCell, #78060), 500 nM A83-01 (StemCell, #72022), 5 μM Y-27632 (Fisher, #72302), and 500 nM SB202190 (Sigma, #S7067). After this initial digestion, the supernatant containing small cell clusters was carefully removed. Fresh digestion medium was then added to the remaining larger tissue chunks for an additional hour to complete the dissociation process. Following digestion, tissue chunks were washed with DMEM/F12 and passed through a pre-wetted 100 μm filter to remove any undigested fragments and debris. The flow-through was then pelleted and resuspended in Geltrex® (Thermo, #A1413201) BME at 4 °C then seeded to form domes in pre-warmed 24-well plates. Plates were inverted and allowed to polymerize at 37 °C for 30 min. Subsequently, 500 μL of pre-warmed complete medium was gently added per 35 μL BME dome to maintain the PDOs. After initial seeding, PDOs were cultured for only 3 days prior to any downstream treatments in order to maintain native TIL populations, ensuring their viability and functionality for subsequent analyses.

### Exosome isolation and characterization
Patient whole blood was collected from noncancerous individuals at the Boston Medical Center Diabetes Clinic following IRB-approved protocols and with informed consent. The blood samples, one from a diabetic donor (HbA1c 9.3%) and one from a ND donor, were used exclusively for exosome isolation and were not associated with the individuals from whom breast tumor tissues were obtained for PDO generation. Blood samples were centrifuged at 16,000 rpm at 4 °C for 30 min to separate plasma, which was then further clarified by additional centrifugation and filtration through a 0.2 μm filter to remove large vesicles or apoptotic bodies. Plasma was

prepared to be platelet-free and isolated within 30 min of the blood collection. Exosomes were isolated using size exclusion chromatography with qEV columns (IZON, product code 1CS-35; RRID:SCR_025764) equipped with spherical beads featuring 35 nm pore sizes, fractionating exosomes ranging from 35 nm to 150 nm in diameter. To minimize experimental variability and focus on the most informative metabolic contrasts, we selected exosome donors through a targeted screening process. The T2D donor, a postmenopausal female selected to mitigate potential estrogen-related effects on exosomal cargo[66–68], was chosen based on having the highest HbA1c level among those not on metabolic drugs, including metformin. The ND donor selected was not taking any medications. Preliminary screening indicated that exosomes from this T2D patient produced the most pronounced differences between the treatment group and negative control. Importantly, previous work has demonstrated that ND exosomes elicit consistent biological effects across donors[9], reducing concerns about variability in the ND condition. Purified exosomes were eluted in DMEM/F12 and stored at 4 °C. Storage of the isolated exosomes was short-term (<24 h post-isolation). Size distribution and concentration of exosomes were measured using a NanoSight NS300 system (Malvern Panalytical). Quality control for the exosomal preparations was performed according to previously published protocols[8,9].

## Exosome treatment
We estimated the number of cells within each PDO sample to normalize exosome treatments. Traditional cell counting methods were not suitable as they would disrupt the architecture of the PDOs; instead, we calculated the total cell number by assuming the PDOs were spherical and determining their volume. This volume was then divided by the average volume of a single cell to estimate the number of cells per PDO. This estimation process was repeated across multiple PDOs to determine an average cell count per PDO, which was then multiplied by the total number of PDOs to obtain the overall cell count for each sample. PDOs were retrieved from BME domes by liquefying the BME at 4 °C for 15 min, followed by centrifugation at 500 rcf for 5 min. Exosomes were isolated from plasma samples of ND and T2D donors, yielding approximately $4.55 \times 10^{11}$ and $6.57 \times 10^{11}$ exosomes per milliliter of plasma, respectively. Exosome recovery was consistent between patient groups. For experimental consistency, exosomes were normalized to a concentration of 5000 particles per PDO cell. PDOs were subsequently seeded to form BME domes into 35 mm glass-bottom FluoroDish® plates (World Precision Instruments) as described above. After polymerization, 2 mL of pre-warmed complete medium, supplemented with the same concentration of exosomes, was gently added to each BME dome to sustain PDO cultures. PDOs were cultured for an additional 3 days to allow for effective exosome uptake and interaction before proceeding to downstream applications.

## Microscopy and imaging of organoids
Morphological assessments of PDOs were conducted via reflectance confocal microscopy using Live-Duo LSM 710 system (Zeiss, Thornwood NY; RRID:SCR_018063) at the BU Microscopy Core, equipped with a Plan-Apochromat 20×/0.8 objective (Carl Zeiss Microscopy GmbH) and 543 nm solid state laser. PDOs were mounted directly onto FluoroDish® glass-bottom dishes, leveraging the natural refractive properties of the tissues to negate the need for staining. Images were captured at a resolution of $1024 \times 1024$ pixels to ensure detailed visualization. Image acquisition was performed at 37 °C in a humidified atmosphere containing 5% $CO_2$, optimal for live-cell imaging. Post-acquisition, images were processed using FIJI software (RRID:SCR_002285)[69]; briefly, brightness and contrast were adjusted before applying automatic thresholding. Morphological parameters, including area, perimeter, and circularity of PDOs calculated via particle analysis. Single cells were excluded by applying size filters greater than 300 μm².

## Single-cell library preparation and sequencing
PDOs were enzymatically digested into single cells using TrypLE Express (Thermo Fisher Scientific) to ensure a gentle yet effective separation. Cells

were resuspended in 0.1% BSA in PBS and passed through a 40 μm cell strainer to achieve a uniform single-cell suspension. Cells were stained with DAPI to identify and exclude non-viable cells and isolated using a FACSAria II cell sorter (BD Biosciences; RRID:SCR_018934) in the BU Flow Cytometry Core. Sorted cells from each treatment group were loaded into individual wells of a 10× Chromium microfluidics chip for single-cell barcoding. Libraries were prepared using the Chromium Single Cell 3′ Library & Gel Bead Kit v3.1 (10× Genomics; RRID:SCR_024537) according to standard protocols. Quality and quantity of the amplified cDNA were assessed using an Angilent Bioanalyzer 2100 High Sensitivity Kit. Libraries were then multiplexed and sequenced on an Illumina NextSeq 2000 platform (RRID:SCR_023614), aiming for a depth of approximately 30,000 reads per cell, which provides sufficient coverage to detect most expressed genes.

## Single-cell data processing, clustering annotation, and integration
Transcriptomic data were mapped to the human reference genome (GRCh38.p14; RRID:SCR_006553) and assigned to individual cells of origin according to cell-specific barcodes using the Cell Ranger pipeline version 6.0.1 (10× Genomics; RRID:SCR_017344). Filtering, feature selection, clustering, and other secondary analyses were performed using Seurat v5.0.1 (RRID:SCR_016341)[70]. Initial preprocessing included the removal of doublets using the DoubletFinder v2.0.4 pipeline (RRID:SCR_018771)[71]. The data were further filtered to exclude cells with high mitochondrial gene expression (>20%) and aberrant unique feature counts (<100, >9000), followed by normalization and identification of highly variable features using scTransform v0.4.1 (RRID:SCR_022146)[71]. Dimensionality reduction was carried out via PCA using 50 dimensions, followed by clustering using the graph-based Louvain method with a resolution of 0.5 to identify distinct cellular populations. Cell types within these clusters were then automatically annotated using the SingleR pipeline (RRID:SCR_023120)[17], referencing an established comprehensive single cell atlas of breast cancer patient samples[18]. To harmonize the data across patients and mitigate batch effects, we performed dataset integration using scVI[14]. This methodology was selected over alternatives based on k-nearest-neighbor batch effect test, integration local inverse Simpson's index, average silhouette width, and principal component regression comparison from single cell integration benchmarking (scIB) analysis[72].

## Identification of neoplastic from normal breast epithelial cells
The integrity of tumor cells within clusters was confirmed by performing inferCNV analysis (RRID:SCR_021140)[73], which identified loss of heterozygosity and copy number alterations characteristic of tumor cells compared to normal immune cells present within PDOs. Epithelial cells were classified into normal or neoplastic categories using a method previously described[18]. Briefly, the mean of squared, scaled inferred changes at each genomic locus was used to calculate a genomic instability score for each cell. For each patient, the top 5% of cells with the highest genomic instability scores were used to generate an average CNV profile. Cells were then correlated with this profile to obtain malignancy scores and plotted accordingly.

Partitioning around medoids clustering was performed using the pamk function in the fpc v2.2.12 package to determine the optimal number of clusters based on silhouette scores. To call normal from tumor cells, a comprehensive grid search over a range of standard deviation multipliers was conducted and applied to both genomic instability and malignancy scores to maximize the average silhouette width. The resulting thresholds were confirmed through visual inspection of scatter plots and histograms of genomic instability and malignancy scores with thresholds clearly demarcated. Thresholds were further validated by comparing expression profiles with cancer-associated genes and pathways, including the COSMIC CGC (RRID:SCR_002260)[15] and GOBP_MAMMARY_EPITHELIAL_ PROLIFERATION (RRID:SCR_002143)[16].

## Differential gene expression and pathway enrichment

Considering that data integration is performed only in the latent space leaving the raw expression profiles untouched, differential expression analysis needs to account for both biological variance and unwanted covariates such as batch effects. To model these technical effects, we performed differential expression analysis using the FindMarkers function in Seurat, using the statistical framework introduced by MAST (RRID:SCR_016340)[74] and setting a latent variable of patient batch. To further explore the biological processes and pathways associated with the differentially expressed genes, we performed functional enrichment analysis using the DAVID Knowledgebase v2024q2 (RRID:SCR_001881)[31]. GSEA was also conducted using the FGSEA v1.26.0 package (RRID:SCR_020938), drawing on gene sets from the Molecular Signatures Database (MSigDB; RRID:SCR_016863)[22]. Enrichment scores were normalized, with permutations set to 50,000 times, using an adjusted $p$-value cut off of 0.05 to filter the significant enrichment results. Expression of specific gene sets of interest were determined by first calculating the average expression score per cell using the AddModuleScore in Seurat, then utilizing a linear mixed effects regression (LMER) model via the lmerTest v3.1.3 package (RRID:SCR_015656) correcting for batch effect.

## Survival analysis

To assess the broader clinical relevance of our findings, we derived a composite gene signature reflecting the net transcriptional effect of diabetes-associated exosomal signaling. This signature was constructed by combining the top 100 differentially expressed genes upregulated in T2D*exo*–treated PDOs and downregulated in ND*exo*–treated PDOs, capturing genes enriched in the T2D-like immune phenotype. We applied this signature to bulk RNA-seq data from two large cancer cohorts: TCGA[25] and METABRIC[26]. Breast cancer cases were subsetted for our analyses. Signature scores were computed for each tumor sample using the mean expression of signature genes normalized per dataset. We used the brcaSurv v0.0.1 package to assess associations between signature scores and overall survival, adjusting for age, PAM50 intrinsic subtype, proliferation score, and inflammation score. Cox proportional hazards models were fit separately for each cohort, and patients were stratified by signature tertiles for visualization using Kaplan–Meier survival curves.

## Signature projection and validation

To validate transcriptional programs observed in the PDO model, we applied gene signature scoring to our recently assembled pan-breast cancer single-cell atlas, comprising 621,200 high-quality cells from 138 patients across 8 independent studies[21]. Cells were annotated as epithelial, stromal, or immune based on canonical markers and clustering, and we quantified the proportion of each compartment per patient. For downstream analysis, we focused on the immune compartment. Using the AddModuleScore function from Seurat, we projected the ChopT signature (derived from our PDO dataset), canonical T cell effector modules, and stress-related gene sets from MSigDB onto individual immune cells. We then computed Spearman correlations between ChopT and comparator module scores to assess co-enrichment of ER stress signatures and suppression of effector function.

## Differential cell type proportion analysis

To compare the abundance of different cell types between experimental conditions in our dataset, we utilized a binomial generalized linear model framework via the lmerTest v3.1.3 package (RRID:SCR_015656) to model the count of cell types across conditions while controlling for covariates such as patient batch. Differential cell type proportion was then quantified as a log odds ratio, reporting the enrichment or depletion of cell types across conditions, using estimated marginal means obtained from the binomial model via the emmeans v1.10.1 package (RRID:SCR_018734).

## miRNA target enrichment

To assess enrichment of predicted and validated miRNA targets among genes downregulated in T2D*exo*-treated LTs, we used a one-sided Fisher's exact test and hypergeometric test framework. Target genes for miR-374a-5p, miR-93-5p, and let-7b-3p were obtained via the multiMiR v1.24.0 package[75], including both validated and high-confidence predicted interactions. Enrichment was tested relative to all expressed genes in the dataset.

## Ligand-receptor analysis

Cell-cell communication within and between tumor and microenvironment populations was analyzed via CellChat v2.1.2 (RRID:SCR_021946)[35] with default configurations. This analysis incorporated a curated database of ligand-receptor interactions, providing a statistical framework for inferring lineage-specific interactions. The T2D*exo*-PDO and ND*exo*-PDO data were extracted and formatted into CellChat format. Data processing and visualization were performed with default settings.

## Pseudotemporal ordering to infer cell trajectories

Diffusion-pseudotime analysis was performed using the Monocle3 v1.3.5 package (RRID:SCR_018685)[29]. Seurat single cell objects containing epithelial cells were converted into a Monocle object using SeuratWrappers v0.3.4 package (RRID:SCR_022555), followed by standard clustering and trajectory inference. The trajectory was rooted in normal LECs to establish the pseudotime ordering. Differential gene expression along the trajectory was assessed using the graph_test function in Monocle3, identifying genes significantly associated with the progression of cell states. Genes were grouped into modules of co-expression via the find_gene_modules function in Monocle3. Module scores were then calculated for each cell using the AddModuleScore function in Seurat.

We also performed pseudotime and hierarchical state analysis on the T cell compartment to investigate immune lineage reprogramming. T cells were first subsetted from the unintegrated Seurat object and reclustered. K2Taxonomer v1.0.7[76] was applied to define a hierarchical taxonomy of T cell states using gene set–level expression features. Pathway enrichment analysis via the runGSEmods function was used to annotate major transcriptional programs across branches. To model T cell differentiation trajectories, we applied Monocle 3 to the same subset. Dimensionality reduction and graph learning were performed using standard parameters, and pseudotime ordering was rooted in Tcm cells, reflecting their early differentiation status. These combined analyses allowed us to capture both lineage structure and underlying pathway signatures associated with dysfunctional T cell fates.

## Statistics and reproducibility

Statistical analyses were performed using R version 4.3.1 (RRID:SCR_001905). Specific tests, significance thresholds, and sample sizes are provided in the figure legends. For organoid experiments, $n$ refers to biological replicates derived from distinct PDO lines. For exosome treatments, replicates represent independent PDOs treated with plasma-derived exosomes from either a ND or T2D donor. All analyses were conducted on biologically independent samples unless otherwise noted.

## Reporting summary

Further information on research design is available in the Nature Portfolio Reporting Summary linked to this article.

## Data availability

All raw and processed single-cell RNA sequencing data will be available from the Gene Expression Omnibus (GEO) under accession number GSE302054 and will be publicly accessible at the time of publication. Metadata for the PDOs is provided in Supplementary Data 1. All other data supporting the findings of this study are available from the corresponding author upon reasonable request.

## Code availability

All original code used in this study will be made publicly available at https://github.com/montilab/BrCaExoPDO/ at the time of publication.

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

## Acknowledgements

We are grateful for David Sherr, Robert Fisher, and Jason Weis for their valuable discussions on data interpretation and manuscript revision, Lina Kroehling for her advice on analysis and statistics, Vinay Duggineni and Christina McConney for their help with cell cluster annotation, and Emma Kelley and Ruben Dries for their support during the setup of the PDO platform. We appreciate the technical assistance and expertise of the BU Flow Cytometry Core Facility, Christopher Williams at the BU Single Cell Sequencing Core, Yuriy Alekseyev at the BU Microarray Sequencing Core, and Michael Kirber at the BU Cellular Imaging Core. We also thank the study nurses and clinical staff that obtained samples and the patients who generously consented for the research use of these samples. Finally, we appreciate members of the Denis and Monti laboratories as well as students and faculty in the BU Virology, Immunology, and Microbiology Department and Immunology Training Program for their feedback and support. This work was supported by grants from NIH: U01CA182898, U01CA243004, and R01CA222170 to G.V. Denis; 5T32AI007309 (PI: Gummurulu) to C.S. Ennis.

## Author contributions

Conception and design: C.S.E., G.V.D. Methodology: C.S.E., S.M., G.V.D. Acquisition of data: C.S.E. M.S., H.K., A.I., K.M., N.Y.K. Analysis and interpretation of data: C.S.E., A.C., S.M. Writing, editing, and revision of manuscript: C.S.E., M.S., A.C., G.V.D. Study supervision and funding: G.V.D.

## Competing interests

The authors declare no competing interests.
