## [Transparent Peer Review file · Communications Biology]

Plasma exosomes from individuals with type 2 diabetes drive breast cancer aggression in patient-derived organoids

Corresponding Author: Professor Gerald Denis

Version 0:

Reviewer comments:

Reviewer #1

(Remarks to the Author)

The study by Ennis, CS et al., aimed to evaluate the effect of exosomes derived from individuals with type 2 diabetes (T2D) on cell type composition and phenotypes in patient-derived breast cancer organoids. In this study, exosomes isolated from plasma from individuals with T2D were cultured with estrogen receptor (ER) positive breast cancer organoids and subsequently subjected to single-cell RNA sequencing analysis. The authors demonstrate that exosomes from T2D impact cell populations and pathways associated with immune suppression and tumor aggressiveness/progression compared to non-diabetic derived exosomes. While this manuscript is of high interest to the research field, this reviewer has a few comments that should be addressed before publication.

Major comments.

1. While it would make sense that female donors were used, it is not indicated in the methods or results whether the plasma donor specimens were derived from males or females. If female donors were used, were the non-diabetic and T2D donors matched for age and menopausal status? Although limited, there is some evidence that estrogen can impact exosome/extracellular vesicle cargo, which could potentially be a confounder of the observed effects. See PMID: 38499957, PMID: 37252969, PMID: 28195143. If donors were not matched, the authors should discuss this potential limitation.
2. In Figure 3, it is unclear why the authors expect upregulation of oncomiR target gene expression for miRNAs that are enriched in T2D-derived exosomes (i.e., miR-374a). Furthermore, did the authors observe a down-regulation of genes responsible for MMP regulation and are they specific targets of the identified oncomiRs?
3. In Figure 6, it would be helpful to the reader/reviewer to indicate the pathways of interest discussed in the results in Figure 6e. It is unclear why the authors focused on these specific pathways over the other 69 pathways.

Minor comments.

1. Lettering is missing from panels in Figure 6.

Reviewer #2

(Remarks to the Author)

This study uses a slightly modified protocol to establish ER+ breast cancer PDO that retain some stromal/immune cell elements. This PDO system is used to interrogate the effects of exosomal signaling from a non-diabetic and a diabetic donor in driving cancer aggressiveness and modulation of the immune system. As presented it is not clear what the main claim of the study is. It is mostly confirming the already known effects of exosomes without additional molecular insight. The single-cell RNAseq analysis mostly provided confirmatory evidence of already known effects.

Major concerns

It is not clear how different this protocol for establishing PDO is or how well it can control the amount of stromal cells that remain in the isolate. From Figure 1 it appears that about 90% of cells represented are epithelial which is an overrepresentation of tumors. The resulting PDOs seem to be rather heterogeneous based on the data from 3 patients Fig 1C. There are concerns that this heterogeneous presentation of the 3 PDOs may be confound result interpretation. PDO318 seems to almost completely devoid of any T, B cells and macrophages. For the two PDOs with more immune cell representation the proportion of epithelial cell subpopulation are vastly different from PDO318.

The interpretation of differential gene expression between exposure to exosomes derived from non-diabetic and diabetic donors are circumscribed to internal comparison. There is no validation with a standard PDO protocol, other patient cohort or dataset such as TCGA. There are concerns that these findings are limited to the PDO models used and it is not clear how this will be generalizable.

The authors acknowledge that their original hypothesis of upregulation of check point ligands on T cells was not supported by their findings in these PDO models. Thus, there is a concern that these PDO may not a bona fide representation of the TME. Without a clear functional validation of what these PDO models represent, it is difficult to appreciate the importance of a posteriori gene expression analysis.

There is little information on the patient characteristics from which PDOs were derived. Similarly, there is little clinical information of the two donors. It is not clear whether other clinical parameters other than diabetes, such as sex, age, BMI, medications may have contributed to the observed gene expression changes.

The authors identified and proposed miR-374a and Notch signal as important mediation of exosomal signaling, but not functional studies or validation are presented.

Minor concerns

Clear identification of each PDO in figures and how (cell proportions, etc) each PDO model contribute to differential gene analysis will be helpful.

Version 1:

Reviewer comments:

Reviewer #1

(Remarks to the Author)

The authors have adequately addressed all of my prior comments.

Minor comment: There are 2 Figure 6's, I assume the second Figure 6 should be "Figure 7".

Reviewer #2

(Remarks to the Author)

The authors have been very responsive to my concerns and suggestions and have sufficiently and satisfactorily addressed them.

the only minor thing for the authors to consider is to more explicitly acknowledge the limited number of PDOs (n = 3) and donors (n =2) as limitation of this study and the need validate independently. This language is provided in the rebuttal letter and also suggest the authors are already expanding this work based on their atlas preprint.

Reviewer #1 (Remarks to the Author):

The study by Ennis, CS et al., aimed to evaluate the effect of exosomes derived from individuals with type 2 diabetes (T2D) on cell type composition and phenotypes in patient-derived breast cancer organoids. In this study, exosomes isolated from plasma from individuals with T2D were cultured with estrogen receptor (ER) positive breast cancer organoids and subsequently subjected to single-cell RNA sequencing analysis. The authors demonstrate that exosomes from T2D impact cell populations and pathways associated with immune suppression and tumor aggressiveness/progression compared to non-diabetic derived exosomes. While this manuscript is of high interest to the research field, this reviewer has a few comments that should be addressed before publication.

Major comments.

- 1. While it would make sense that female donors were used, it is not indicated in the methods or results whether the plasma donor specimens were derived from males or females. If female donors were used, were the non-diabetic and T2D donors matched for age and menopausal status? Although limited, there is some evidence that estrogen can impact exosome/extracellular vesicle cargo, which could potentially be a confounder of the observed effects. See PMID: 38499957, PMID: 37252969, PMID: 28195143. If donors were not matched, the authors should discuss this potential limitation.*

We thank the reviewer for this insightful comment. For this study, exosomes were derived from plasma from females: a single non-diabetic (ND) donor and a single type 2 diabetic (T2D) donor. This one-to-one design ensures that any observed differences between ND- and T2D-exosome-treated PDOs are primarily driven by exosome treatment rather than donor variability.

While detailed clinical histories on the ND donor were not available, our prior work demonstrates that ND exosomes exhibit a high degree of biological reproducibility and inter-donor similarity. Specifically, in prior work analyzing prostate cancer cell lines treated with exosomes, principal component analysis (PCA) of whole-genome RNA sequencing data showed that male ND-exosome-treated samples clustered tightly together, demonstrating a consistent biological effect across ND donors. Additionally, ND-exosome-treated samples *clustered with untreated controls*. These signals also remained clearly distinct from T2D-treated samples, further supporting their similarity to each other and robust reproducibility (PMID: 36644690, Fig. S10A from that publication). Given this high degree of similarity, additional ND donors would be expected to yield highly consistent results, making the use of a single ND donor, despite limited clinical information, a reasonable and well-controlled approach for this study.

To further reduce potential estrogen-related confounding, we specifically selected a female, postmenopausal T2D donor. Estrogen has been reported to influence extracellular vesicle cargo; however, since our selected T2D donor was postmenopausal, any estrogenic impact on exosome composition would be expected to be minimal. We have added clarifying details about donor selection in the Methods section (starting at line 512).

Recognizing that medications can influence exosome content, we carefully selected a T2D donor with minimal medication exposure (only insulin and a non-glucose-lowering medication), ensuring that observed effects primarily reflect diabetes status rather than pharmacological influences. While a comprehensive evaluation of medication effects on exosome biology is well beyond the scope of this study, we recognize this feature as an important future research direction to better understand how metabolic interventions shape exosome function and tumor interactions. Future studies could also potentially investigate the role of aerobic exercise, diet, age and race in ND donors. There is a rationale to pursue this direction in future studies, because some of our recent work (PMID: 40047645) suggests that exosomes isolated from the insulin sensitive/ND physiological state may contain anti-metastatic factors.

- 2. In Figure 3, it is unclear why the authors expect upregulation of oncomiR target gene expression for miRNAs that are enriched in T2D-derived exosomes (i.e., miR-374a). Furthermore, did the authors observe a down-regulation of genes responsible for MMP regulation and are they specific targets of the identified oncomiRs?*

We thank the reviewer for this important question and the opportunity to clarify our interpretation. In our initial GSEA of LT1 marker genes—a cluster expanded in response to T2D_{Dexo} treatment—we observed enrichment for gene sets composed of predicted targets of T2D-associated oncomiRs (miR-374a-5p). While this might suggest upregulation of miRNA targets, GSEA identifies enrichment within a ranked gene list and does not imply directionality or direct miRNA-mediated repression. Upon closer examination, LT1 markers were also enriched for gene sets related to chromatin remodeling and vesicle trafficking and endocytosis. These features suggest that LT1 cells may be particularly poised to internalize exosomes and undergo transcriptional reprogramming in response to their molecular cargo.

Prompted by this comment, we more directly assessed the impact of T2D_{Dexo}-derived miRNAs by performing a targeted enrichment analysis comparing differentially expressed genes in LTs treated with T2D_{Dexo} versus ND_{Dexo}. This analysis revealed highly significant enrichment of validated and predicted miRNA targets among **downregulated** genes (Fisher's exact test, $p < 2.2 \times 10^{-16}$; odds ratio = 3.6), consistent with canonical miRNA-mediated repression and confirming functional activity of these miRNAs in recipient tumor cells.

In relation to MMP regulation, we observed downregulation of key negative regulators—including *TIMP2* and *LACTB*—specifically within T2D_{Dexo}-treated LT clusters. *TIMP2* is a predicted target of miR-93-5p, and *LACTB* is a validated target of miR-374a-5p (multiMiR; PMID: 29790671). These findings suggest that T2D-derived oncomiRs may enhance MMP pathway activity by repressing inhibitory regulators, rather than directly upregulating MMP transcripts, in line with our previous reports (PMID: 34813359).

We have updated the Figure 3 and Results section (beginning at line 216) to reflect this clarified interpretation and have moved the LT1-specific GSEA findings to the Supplement, where they remain as exploratory observations relevant to the distinct transcriptional state of this cluster. We have also incorporated the miR prediction results in the Results (Fig. 3E-F; beginning at line 225) and Methods (beginning at line 654).

3. In Figure 6, it would be helpful to the reader/reviewer to indicate the pathways of interest discussed in the results in Figure 6e. It is unclear why the authors focused on these specific pathways over the other 69 pathways.

We thank the reviewer for this helpful comment. As noted in the manuscript, all pathways presented in Figure 6e are statistically significant, including 69 conserved, 10 ND_{Dexo}-specific, and 16 T2D_{Dexo}-specific pathways. In the Results section, we focused on a subset of pathways that were most biologically relevant to our central hypothesis: the role of diabetes-associated exosomes in modulating the tumor immune microenvironment and promoting aggressive signaling.

Specifically, we highlighted immunoregulatory (e.g., MHC-I/II, CXCL) and pro-tumorigenic signaling pathways (e.g., NOTCH, BMP, MMP), which reflect core themes of immune modulation and tumor progression. These pathways were selected based on prior literature, our own functional studies (PMID: 34813359, PMID: 40047645), and consistency with previously reported mechanisms in T2D-related cancer biology. To address the reviewer's concern and improve interpretability, we have visually distinguished the discussed pathways in the

figure and clarified this rationale in the manuscript (starting at line 324). This will help guide the reader while preserving the full view of the enriched pathway landscape. Please note that this figure is now presented as **Figure 7**, reflecting reordering and additions made in response to other reviewer comments.

Minor comments.

1. *Lettering is missing from panels in Figure 6.*

We appreciate the reviewer for noting this oversight. In the revised manuscript, we have added appropriate panel lettering to all components of the figure. Please note that this figure is now presented as **Figure 7**, reflecting reordering and additions made in response to other reviewer comments.

Reviewer #2 (Remarks to the Author):

This study uses a slightly modified protocol to establish ER+ breast cancer PDO that retain some stromal/immune cell elements. This PDO system is used to interrogate the effects of exosomal signaling from a non-diabetic and a diabetic donor in driving cancer aggressiveness and modulation of the immune system. As presented it is not clear what the main claim of the study is. It is mostly confirming the already known effects of exosomes without additional molecular insight. The single-cell RNAseq analysis mostly provided confirmatory evidence of already known effects.

Major concerns

- 1. It is not clear how different this protocol for establishing PDO is or how well it can control the amount of stromal cells that remain in the isolate. From Figure 1 it appears that about 90% of cells represented are epithelial which is an overrepresentation of tumors. The resulting PDOs seem to be rather heterogeneous based on the data from 3 patients Fig 1C. There are concerns that this heterogeneous presentation of the 3 PDOs may be confounding result interpretation. PDO318 seems to almost completely be devoided of any T, B cells and macrophages. For the two PDOs with more immune cell representation the proportion of epithelial cell subpopulation are vastly different from PDO318.

Thank you for your insightful comments regarding the cellular composition and heterogeneity of our PDOs. We acknowledge that direct comparisons to standard PDO protocols are challenging, as existing methods do not retain tumor-infiltrating lymphocytes (TILs). A key innovation of our approach is the optimized short-term culture and dissociation technique, which overcomes this limitation by preserving tumor, stromal, and immune cell populations. This allows for a more comprehensive and physiologically relevant analysis of the tumor microenvironment (TME), providing a critical platform for studying immune-tumor interactions that are otherwise lost in conventional PDO models.

To contextualize the epithelial-stromal-immune composition in our PDOs, we refer to our group’s recently assembled largest-to-date atlas of breast cancer single-cell studies, which includes 621,200 high-quality single cells from 138 patients across 8 independent studies (preprint on *bioRxiv*; DOI: 10.1101/2025.03.13.643025). This resource reveals substantial heterogeneity in the single-cell landscape across these datasets. Some tumors are composed almost entirely of epithelial cells (>99%), while others are predominantly stromal- or immune-rich (>99%). Given this broad spectrum of variation in primary tumors, the heterogeneity observed in our PDOs is consistent with real patient tumor samples rather than a confounding artifact of our protocol.

The differences in immune cell representation across PDOs, including the lower immune content in PDO318, likely reflect interpatient variability in tumor-immune composition, which has been widely documented in breast cancer (PMID: 30193111, PMID: 34493872, PMID: 26330355). While this variability is expected, we explicitly account for it in our analyses by including patient identity as a latent variable in the statistical framework of our differential expression and proportion analyses.

To further support these claims, we have appended Figure S1E comparing our PDO data to the broader cell-type distribution in our breast cancer single-cell atlas (preprint on bioRxiv; DOI: 10.1101/2025.03.13.643025). This benchmarking analysis demonstrates that the epithelial and immune cell proportions in our PDOs fall within the expected range of variation observed in primary tumors. We have also revised the main text of the manuscript accordingly (beginning at line 132) to clarify these points and guide the reader through this rationale. Please note that the atlas is currently under review elsewhere and is provided here for confidential reviewer evaluation only.

2. *The interpretation of differential gene expression between exposure to exosomes derived from non-diabetic and diabetic donors are circumscribed to internal comparison. There is no validation with a standard PDO protocol, other patient cohort or dataset such as TCGA. There are concerns that these findings are limited to the PDO models used and it is not clear how this will be generalizable.*

We thank the reviewer for this insightful comment. We acknowledge the importance of validating our findings beyond the internal comparison of exosome exposure in our PDO models.

To assess the broader relevance of our findings, we constructed a composite gene signature comprising the top differentially expressed genes from both the T2D-like and ND-like conditions (i.e., genes upregulated in T2D-exosome-treated PDOs and those downregulated in ND-exosome-treated PDOs). This signature captures the net transcriptional effect of diabetes-associated exosomal signaling. We applied this composite signature to both the TCGA and METABRIC breast cancer cohort and evaluated its association with overall survival, adjusting for age, proliferation score, inflammation score, and PAM50 intrinsic subtype. The signature was significantly associated with worse survival (TCGA: HR = 2.23, $p < 0.002$; METABRIC: HR = 2.06, $p < 1.2e-05$), underscoring its prognostic value in an independent patient dataset and supporting the generalizability of our findings.

To interpret the biology underlying this signature, we performed separate functional annotations of the T2D-like and ND-like gene sets used to construct the composite. The T2D-like signature revealed a diverse, pro-tumorigenic transcriptional program: 26.47% metabolic, 19.12% transcription factors/signaling, 16.18% invasion, proliferation, or EMT-related, 11.76% ECM/structural, 11.76% sensory, 7.35% stemness/developmental, and 5.88% immune-related, half of which were regulatory or suppressive in nature. In contrast, the ND-like signature was overwhelmingly immune-enriched (91.40%), supporting its association with a protective, anti-tumor immune phenotype. Further analysis of this gene set showed that 30.59% of immune-related genes were involved in TCR signaling or T cell activation, 18.82% in leukocyte migration, and 16.47% in T cell development—a pattern consistent with productive anti-tumor immune responses. These findings are consistent with clinical and epidemiological studies reporting that diabetic women experience worse breast cancer outcomes (PMID: 29141994, PMID: 31720917), and they underscore the broader relevance of diabetes-associated transcriptional programs across patient datasets. To incorporate these points, we have revised the main text starting at line 166 to include this analysis and its clinical significance, along with the supporting hazard ratio data and signature interpretation (Fig. 2g, Table S6). We have also updated our Methods section to incorporate the details of these new analyses (line 635).

However, we note that TCGA does not include metabolic status information, which presents a limitation in directly assessing diabetes-associated tumor biology at scale. This highlights a critical gap in publicly available datasets and underscores the importance of our study in pushing the field forward by systematically integrating metabolic status into tumor biology research. While TCGA provides an independent dataset for validation, additional prospective studies incorporating metabolic status would further strengthen these findings, though they are beyond the scope of this current work.

Regarding comparisons to standard PDO protocols, such validation is not feasible due to fundamental methodological differences. Traditional PDOs lack TILs (PMID: 32654925), owing to extended culture times and digestion methods that deplete immune cells. In contrast, our protocol preserves both tumor and immune compartments by using short-term culture and optimized dissociation, allowing for integrated analysis of tumor-immune dynamics. We believe this innovation enables a more physiologically relevant model of the TME and provides a valuable platform for understanding how metabolic disease influences tumor-immune interactions.

- 3. The authors acknowledge that their original hypothesis of upregulation of check point ligands on T cells was not supported by their findings in these PDO models. Thus, there is a concern that these PDO may not a bona fide representation of the TME. Without a clear functional validation of what these PDO models represent, it is difficult to appreciate the importance of a posteriori gene expression analysis.*

We thank the reviewer for this thoughtful and important comment. While our original hypothesis — that checkpoint ligands would be upregulated in response to T2D-derived exosomes — was not supported, we interpret this as biologically meaningful rather than a limitation of the PDO model. Specifically, our data point to a non-canonical, checkpoint-independent form of T cell dysfunction, reflective of the immunometabolic alterations introduced by T2D.

ER stress is a well-established mechanism of T cell dysfunction (PMID: 38297380, PMID: 36755160, 33214692), particularly in chronically stimulated or tumor-infiltrating contexts. Activation of the unfolded protein response (UPR) under conditions of metabolic stress, hypoxia, or persistent antigen exposure leads to the upregulation of transcription factors such as CHOP (*DDIT3*), a downstream effector of the PERK–ATF4 axis. CHOP acts as a key mediator of ER stress–induced dysfunction by suppressing T-bet (*TBX21*) and thereby limiting the expression of effector molecules such as IFN- γ and granzyme B (PMID: 30894532, PMID: 30305738). Studies have demonstrated that CHOP expression correlates with mitochondrial dysfunction, apoptosis, and loss of memory potential in T cells from both tumors and chronic infections (PMID: 30659052, PMID: 36919984). Importantly, pharmacologic or genetic inhibition of CHOP has been shown to reinvigorate T cell function (PMID: 30894532), highlighting ER stress not only as a marker but also as a barrier to immune efficacy.

In parallel, ER stress plays a central role in T2D pathophysiology, contributing to β -cell failure, insulin resistance, and chronic inflammation (PMID: 18048764, PMID: 21233852, PMID: 22443930, PMID: 25493331). However, to our knowledge, this is the first study to mechanistically link ER-stressed T cells with diabetes-associated tumor microenvironments. Rather than undermining the fidelity of our PDO system, these findings underscore its utility in modeling the complex immunometabolic crosstalk characteristic of comorbid conditions like cancer and T2D.

Importantly, immune evasion via PD-1/PD-L1 has been primarily studied in metabolically healthy settings. Our findings suggest that T2D fundamentally reshapes the tumor–immune interface, such that checkpoint signaling may no longer be the dominant immunosuppressive mechanism. This aligns with clinical data showing that T2D patients exhibit reduced responsiveness to checkpoint blockade (PMID: 31720917), supporting the need for alternative therapeutic strategies targeting metabolic and stress pathways.

Prompted by this comment, we conducted additional analyses to further define the T cell phenotype in our model. Pseudotime trajectory analysis revealed that CD8⁺ T cells in T2Dexo-PDOs diverged from canonical activation programs, instead following a path marked by ER and metabolic stress, downregulation of effector molecules (*IL7R*, *GZMK*, *TBX21*), and upregulation of *CHOP*, *XBPI*, and the immunoregulatory enzyme *IDO1* — collectively supporting an alternative, checkpoint-independent suppressive state.

To validate this finding in a broader clinical context, we applied the ChopT signature (defined from our PDO dataset) to our recently assembled pan-breast cancer single-cell atlas, which includes 621,200 high-quality

immune cells from 138 patients across 8 independent studies. Using Seurat’s AddModuleScore, we scored each cell for ChopT activity, a curated effector T cell module, and multiple stress-related gene sets from MSigDB. We observed statistically significant negative correlations between ChopT and effector-related programs (e.g., $\rho = -0.24$ for effector score, $\rho = -0.14$ for cytotoxicity score, $\rho = -0.12$ for IFNG signaling; $p < 1e-10$) and positive correlations with stress-related pathways (e.g., $\rho = 0.34$ for hypoxia score; $\rho = 0.30$ for UPR score; $\rho = 0.24$ for ER stress apoptosis score; $p < 1e-10$). These results confirm that CHOP-high T cells are functionally impaired and stress-enriched across diverse clinical settings.

Although our analysis focuses on transcriptomic signatures, the observed upregulation of stress response genes (*DDIT3*, *XBP1*) and downregulation of effector regulators (e.g., *TBX21*, *GZMK*) are functionally well-validated markers of impaired cytotoxic T cell activity. We have clarified these interpretations in the revised text (starting at line 422).

Crucially, this phenotype was specific to T cells in T2D-treated PDOs and was not observed across other immune populations or in normoglycemic controls, ruling out a global stress artifact. These findings support a model in which ER and metabolic stress precede or bypass classical checkpoint exhaustion, consistent with recent studies implicating CHOP and XBP1 in exhaustion-resistant dysfunctional states (e.g., PMID: 30894532, PMID: 39845962).

We have incorporated these new analyses and interpretations into the revised manuscript (Fig. 6; Results section starting at line 290). We have also updated our Methods section to incorporate the details of these new analyses (line 677 for pseudotime and line 635 for atlas projection).

4. *There is little information on the patient characteristics from which PDOs were derived. Similarly, there is little clinical information of the two donors. It is not clear whether other clinical parameters other than diabetes, such as sex, age, BMI, medications may have contributed to the observed gene expression changes.*

We appreciate the reviewer’s comment and the opportunity to clarify these details. Patient characteristics for PDO generation, including age, sex, and tumor characteristics, are provided in Supplementary Table 1. The two exosome donors (one ND and one T2D) were cancer-free and stratified solely based on metabolic status. While detailed clinical information such as BMI and medication history was not available for the ND donor, our prior work has shown that exosomes from nondiabetic individuals exert consistent effects across donors, minimizing concerns about donor variability (see response to Reviewer 1).

We recognize that obesity-associated cancers involve considerable clinical heterogeneity—including variations in age, metabolic state, and comorbidities. Indeed, this complexity has historically hindered research

and clinical trials, with T2D and its associated organ dysfunction (e.g., cardiovascular disease) often serving as exclusion criteria and thus limiting our ability to understand how metabolic dysfunction interacts with tumor biology. Our work takes a critical first step toward modeling this complexity by systematically investigating the impact of T2D status on the immune microenvironment.

Moreover, our previous studies indicate that T2D status exerts a more pronounced effect on immune-related pathways in the TME than other metabolic factors such as cholesterol levels or waist-to-hip ratio (PMID: 25012997, PMID: 23068072, PMID: 29141994, PMID: 27995352, PMID: 26900131). These findings are supported by extensive evidence linking T2D to immune dysregulation (PMID: 29738558, PMID: 31657690, PMID: 28408379), reinforcing the rationale for focusing on diabetes-driven immune remodeling in this study.

We acknowledge the complexity of this field, and future studies incorporating larger and more diverse patient cohorts with expanded clinical profiling will be essential to further dissect the interplay between metabolic dysfunction and tumor immunity.

5. *The authors identified and proposed miR-374a and Notch signal as important mediation of exosomal signaling, but not functional studies or validation are presented.*

We appreciate the reviewer's comment. Both miR-374a and Notch signaling have been functionally validated in our prior work. Specifically, our previous study demonstrated Notch pathway activation in MCF7 cell lines treated with T2D adipocyte-derived exosomes (PMID: 34813359). Additionally, we identified miR-374a as differentially upregulated in T2D vs. ND plasma exosomes, with miR-374a transfection driving upregulation of aggressive gene expression programs in DU145 cell lines (PMID: 36644690).

Our current findings further support these mechanisms while extending our previous conclusions to a more physiologically relevant model. Unlike our prior work in immortalized cell lines, this study investigates miR-374a and Notch signaling within PDOs, which better recapitulate tumor heterogeneity and immune interactions within the diabetic TME. Given that these pathways have already been functionally confirmed, our study demonstrates their relevance within a PDO model that more accurately reflects patient tumors.

Minor concerns

1. *Clear identification of each PDO in figures and how (cell proportions, etc) each PDO model contribute to differential gene analysis will be helpful.*

We thank the reviewer for this comment. We agree that clearly identifying each PDO and its contribution to differential gene expression analysis is important for interpretation.

As presented in Figure 1C, we have included cell proportion data per PDO line, providing a clear breakdown of epithelial, stromal, and immune cell distributions across PDOs. Additionally, Supplementary Figure 1A presents a UMAP visualization colored by patient identity, demonstrating interpatient variability in global gene expression patterns.

In all other figures, data from all patients are analyzed collectively. Pooling data from all PDOs allows for increased statistical power and a more comprehensive representation of tumor heterogeneity across patients, which is a widely used approach in single-cell transcriptomic analyses. To ensure that patient-specific differences do not confound our differential expression and proportion analyses, we included patient batch as a covariate in all models. This statistical approach effectively controls for interpatient variability and ensures that differentially expressed genes reflect true biological differences rather than technical or patient-specific artifacts. We have updated the Results (starting at line 137) and Methods section (starting at line 650) to clarify our approach.